# Planar Seismic Source Characterization Models Developed for Probabilistic Seismic Hazard Assessment of Istanbul

Zeynep Gülerce[1], Kadir Buğra Soyman[1], Barış Güner[2], and Nuretdin Kaymakci[3]

[1]Department of Civil Engineering, Middle East Technical University, Ankara, 06800, Turkey
[2] Department of Nuclear Safety, Turkish Atomic Energy Authority, Ankara, 06510, Turkey
[3] Department of Geological Engineering, Middle East Technical University, Ankara, 06800, Turkey

*Correspondence to*: Zeynep Gülerce (zyilmaz@metu.edu.tr)

**Abstract.** This contribution provides an updated planar seismic source characterization (SSC) model to be used in the probabilistic seismic hazard assessment (PSHA) for Istanbul. It defines planar rupture systems for the four main segments of North Anatolian Fault Zone (NAFZ) that are critical for the PSHA of Istanbul: segments covering the rupture zones of 1999 Kocaeli and Düzce earthquakes, Central Marmara, and Ganos/Saros segments. In each rupture system, the source geometry is defined in terms of fault length, fault width, fault plane attitude, and segmentation points. Activity rates and the magnitude recurrence models for each rupture system are established by considering geological and geodetic constraints and are tested based on the observed seismicity that is associated with the rupture system. Uncertainty in the SSC model parameters (e.g. b-value, maximum magnitude, slip rate, weights of the rupture scenarios) is considered whereas; the uncertainty in the fault geometry is not included in the logic tree. To acknowledge the effect of earthquakes that are not associated with the defined rupture systems on the hazard, a background zone is introduced and the seismicity rates in the background zone are calculated using smoothed-seismicity approach. The state-of-the-art SSC model presented here is the first fully-documented and ready-to-use fault-based SSC model developed for the PSHA of Istanbul.

## 1 Introduction

North Anatolian Fault Zone (NAFZ), one of the most active fault systems in the world, extends for more than 1500 kilometers along Northern Turkey (Figure 1b). NAFZ was ruptured progressively by eight large and destructive earthquakes ($M_w$>6.5) in the last century. Earthquakes that occurred between 1939 and 1967 had ruptured approximately 900 kilometers of a uniform trace in the east whereas; the 1999 Kocaeli and Düzce Earthquakes ruptured a total fault span of approximately 200 kilometers where the NAFZ is divided into a number of branches in the west. Northern strand of NAFZ is submerged beneath the Marmara Sea to the west of the 1999 Kocaeli Earthquake rupture zone, introducing major uncertainties to segment location, continuity, and earthquake recurrence (Figure 1a). In 2004, Parsons compiledollected a catalog of large magnitude (M>7) earthquakes occurred around the Marmara Sea for the time period of A.D. 1500 - 2000. Based on the rupture zones of these large magnitude events, four main segments for the northern strand of NAFZ around Marmara Sea were proposed by Parsons (2004): (1) Ganos segment that combines the rupture zones of August 1776 and 1912 earthquakes,

(2) Prince Island's segment that includes the rupture zones of 1509 and May 1766 earthquakes, (3) Izmit Segment defined for the rupture zones of 1719 and 1999 earthquakes, and (4) Çınarcık Segment defined for M~7 floating earthquakes (independent normal-fault earthquakes that may have occurred on different fault segments in or around the Çınarcık basin). Parsons (2004) noted that the 10 May 1556 (Ms=7.1), 2 September 1754 (M = 7.0), and 10 July 1894 (M = 7.0) earthquakes were assigned locations in the Çınarcık basin or on mapped normal faults in the southern parts of Marmara Sea. These events were not allocated to the other segments in order not to violate the inter-event time calculations, although they could have occurred on the northern strand of NAFZ.

The fault segmentation model proposed by Erdik et al. (2004) was similar to the segmentation model proposed by Parsons (2004) in terms of the fault geometry; however, smaller segments were preferred. Erdik et al. (2004) noted that "*the Main Marmara fault cuts through Çınarcık, Central and Tekirdağ basins, follows the northern margin of the basin when going through the Çınarcık trough in the northwesterly direction, makes a westwards kink around south of Yeşilkoy until it reaches the 1912 Murefte–Şarköy rupture*". All of these fault lines were interpreted as separate fault segments in the segmentation model. Erdik et al. (2004) considered multi-segment ruptures by assigning lower probabilities to "*cascading ruptures*". Based on the rupture zones of previous large magnitude events, multi-segment ruptures involving the segments in connection with the 1999 Kocaeli earthquake and 1509 earthquake were included in the rupture forecast. Even though multi-segment ruptures were considered, the relative probabilities of the multi-segment ruptures vs. single-segment ruptures were not systematically defined in Erdik et al. (2004). This seismic source model was updated for the Earthquake Hazard Assessment for Istanbul Project by OYO (2007). The fundamental differences between the Erdik et al. (2004) and OYO-2007 models are: (1) small segments around Marmara Sea used in Erdik et al. (2004) model were combined to form bigger segments in OYO-2007 model; (2) fault segments that represent the floating earthquakes were defined. The segmentation model used in OYO-2007 source characterization is very similar to the segmentation model proposed by Parsons (2004).

The fault segmentation model used by Kalkan et al. (2009) includes significant differences in terms of the fault geometry with the Erdik et al. (2004) model, even though both studies used the active fault maps of Şaroğlu et al. (1992) for inland faults and the fault segmentation model from Le Pichon et al. (2003) and Armijo et al. (2005) for the segments beneath the Sea of Marmara. On the other hand, the magnitude recurrence models used by Erdik et al. (2004), in OYO-2007 model, and by Kalkan et al. (2009) were rather similar. In all of these studies, linear fault segments were modeled (fully or partially) by the characteristic model proposed by Schwartz and Coppersmith (1984); therefore, only large magnitude events were associated with the fault segments. Additionally, a background source representing the small-to-moderate magnitude earthquakes (earthquakes between 5 and 6.5-7 depending on the study) were added to the source model and the earthquake recurrence of the background source was modeled using a double-truncated exponential magnitude distribution model. Either the Poisson (Erdik et al., 2004; Kalkan et al., 2009) or time dependent renewal (Brownian Passage Time, Ellsworth et al., 1999) model (Erdik et al. 2004) was chosen to model the earthquake recurrence rates for linear segments; whereas the Poisson distribution was used to model the recurrence rates of the background source in these studies.

Recently proposed SSC models for the western segments of NAFZ (Gülerce and Ocak, 2013 and Murru et al., 2016) are more ~~complicated~~ detailed in terms of the segmentation models, magnitude recurrence relations, and estimation of the activity rates. In the Gülerce and Ocak (2013) SSC model, length of segments and the segmentation points were determined and incorporated with the help of then available fault maps and traced source lines on the satellite images. Planar fault segments were defined and the composite magnitude distribution model (Youngs and Coppersmith, 1985) was used for all seismic sources in the region to properly represent the characteristic behavior of NAFZ without an additional background zone. Unfortunately, the seismic source model proposed by Gülerce and Ocak (2013) cannot be directly implemented in the PSHA for Istanbul since the model does not include the fault segments on the west of 1999 Kocaeli Earthquake rupture zone. Geometry of the fault segments defined in Murru et al. (2016) is generally similar to the Erdik et al. (2004) model. Furthermore, Murru et al. (2016) provided~~; however,~~ the complete set of parameters required for a fault-based PSHA analysis (e.g. slip rates, fault widths, rupture models and rates, parameter uncertainties, etc.) ~~was provided in Murru et al. (2016)~~.

The objective of this study is to provide an updated and properly documented fault-based SSC model to be used in the PSHA studies in Istanbul. A significant portion of the tectonic database is acquired from the Updated Active Fault Maps of Turkey that was published by General Directorate of Mineral Research and Exploration (Emre et al., 2013) (accessed through http://www.mta.gov.tr/v3.0/hizmetler/yenilenmis-diri-fay-haritalari). The 1/250.000 scale Çanakkale (NK 35-10b), Bandırma (NK 35-11b), Bursa (NK 35-12), Adapazarı (NK 36-13), Bolu (NK 36-14), and Istanbul (NK 35-9) sheets of Updated Active Fault Maps of Turkey were accessed and digitized. The seismological database is taken from the Integrated and Homogeneous Turkish Earthquake Catalog published by Kandilli Observatory and Earthquake Research Institute (Kalafat et al., 2011). Seismotectonic information related to the active faults and the fault systems that are available in these databases and in the current scientific literature are used in combination with the segmentation models proposed by Gülerce and Ocak (2013) and Murru et al. (2016) to define the rupture systems. Fault segments, rupture sources, rupture scenarios, and fault rupture models are determined using the terminology given in Working Group of California Earthquake Probabilities (WGCEP-2003) report and multi-segment rupture scenarios are considered in a systematic manner. Events in the seismological database are attributed to the rupture systems and the logic tree weights for the rupture scenarios are determined by comparing the accumulated seismic moment due to the geological constraints (rupture dimensions and slip rate) with the seismic-moment release due to associated seismicity. Different than the previous efforts, the PSHA inputs (e.g. coordinates of the fault segments, logic tree branches and corresponding weights) are properly documented; therefore, the SSC model presented here can be directly implemented in the future site-specific PSHA studies in Istanbul.

**2 Fault Segmentation Models, Rupture Systems, and Partitioning of Slip Rates**

The SSC model consists of one background source (defined in Section 5) and four distinct (non-overlapping) rupture systems that are defined by considering the rupture zones of previous large magnitude earthquakes documented by Parsons

(2004) on the northern strand of NAFZ. We note that all sub-segments in the defined rupture systems except for North and South Çınarcık Segments are assumed to be near vertical with right-lateral slip as suggested by geological, seismological, and GPS data. The segmentation and the slip rate partitioning models are not yet well-established for the fault segments on the south of the Marmara Sea; therefore, these segments are not modelled as planar seismic sources in this SSC model.

## 2.1 Izmit and Düzce Rupture Systems:

Location, geometry, and slip distribution of the rupture zones of 1999 Kocaeli and Düzce earthquakes have been studied extensively after these events (e.g. Barka et al., 2002; Langridge et al., 2002; Akyüz et al., 2002). The surface rupture of the 1999 Kocaeli earthquake extended for almost 165 km and 4 distinct segments were ruptured (Hersek Segment, Gölcük-Karamürsel-Izmit Segment, Sapanca-Akyazı Segment, and Karadere Segment as given in Barka et al., 2002). The co-seismic fault was terminated at the western end of the rupture, very near to the eastern side of the Marmara Sea (Ergintav et al., 2014). The northern strand of NAFZ that delimits the boundary between the Marmara Sea and Çınarcık Block did not rupture during 1999 Kocaeli Earthquake (Çınarcık Segment in Figure 1a). Mert et al. (2016) argued that the northern strand of NAFZ is observed as a single continuous fault strand along Izmit Bay and at its entrance to the sea southeast of Istanbul. We included the North Çınarcık segment (Segment 3) in the Izmit rupture system because it is the western extension of the Hersek-Gölcük Segment that was developed in response to the bending of the main strand of the NAFZ towards NW. This bending results in a releasing bend and a slip re-distribution as dextral motion parallel to the main strand and normal motion perpendicular to the Çınarcık Segments (Figure 1d). As seen in Figure 1e, the vertical throw of the Northern Çınarcık Segment is almost twice of the throw of the South Çınarcık Segment, which is the conjugate fault of the North Çınarcık segment. The dip of the North Çınarcık Segment is assumed to be 70°SW as suggested by Laigle et al. (2008) while the dip of sSouth Çınarcık Segment is assumed to be 60°NW. The Izmit rupture system proposed here consists of five (Hersek-Gölcük, Izmit, Sapanca-Akyazı, Karadare and North Çınarcık) sub-segments. Düzce Earthquake produced 40-km-long surface rupture zone; however, there is a 4-km releasing step-over around Eften Lake (Akyüz et al. 2002). Therefore, a 2-segment model (Segments D1 and D2) is established for the rupture zone of the Düzce earthquake (Figure 1a). The segments and segment lengths for the Izmit and Düzce rupture systems are given in Table 1. In 1999 earthquakes, these two rupture systems (Kocaeli and Düzce) were ruptured in two different episodes. A possible explanation of the separate ruptures in different episodes would be the development of the restraining bend along Karadere Segment, which probably locked up the eastern termination of Izmit rupture. Harris et al. (2002) proposed that the rupture of 1999 İzmit earthquake was stopped by a step- over at its eastern end (Mignan et al., 2015). In this study, we assumed the same rupture pattern of 1999 earthquakes and do not include a rupture scenario that combines these two rupture systems in the rupture forecast.

## 2.2 Ganos/Saros Rupture System:

The ENE-WSW trending Ganos Fault is the fault segment at the westernmost section of NAFZ that generated the 9 August 1912 Mürefte (Ganos) earthquake. Magnitude of this earthquake was estimated from historical catalogues and field

observations as $M_s = 7.3 \pm 0.3$ (by Ambraseys and Jackson, 2000) and $M_w=7.4$ (by Altunel et al., 2004), respectively (Aksoy et al., 2010). A second large event was occurred on 13 September 1912 ($M_s=6.8 \pm 0.35$ and the estimated seismic moment was $2.19 \times 10^{19}$ Nm as given in Ambraseys and Jackson, 2000). Ambraseys and Jackson (2000) suggested a 37-km-long co-seismic rupture for this large second shock. Aksoy et al. (2010) used the duration of the recorded waveforms to estimate the rupture lengths of 1912 events: assuming the rupture width as 15-20 km, estimated values were $130 \pm 15$ km and $110 \pm 30$ km for August 9 and September 13 events, respectively. According to Aksoy et al. (2010), co-seismic surface ruptures were visible along the 45 km on-land section of this segment. Supporting the estimations based on waveforms by aerial photographs, satellite imagery, digital elevation models, bathymetry, and field measurements; Aksoy et al. (2010) proposed $120 \pm 30$ km-long fault rupture for the August 9, 1912 event. Murru et al. (2016) defined two segments covering the $120 \pm 30$ km long fault rupture of the 1912 Ganos Earthquake: a 74 km-long segment that includes the on-land section and a 46 km-long off-shore segment (Segments 6 and 7 in Figure 1a). The maximum seismogenic depth of these segments was assumed to be 15 km on the basis of the locking depth suggested by mechanical best fit modelling of GPS data (Flerit et al., 2003) and by the depth extent of instrumental seismicity (Gürbüz et al., 2000; Özalaybey et al., 2002; Örgülü and Aktar, 2001; Pınar et al., 2003). A similar segmentation model is adopted in this study by implementing minor changes in the sub-segment lengths as shown in Table 1.

## 2.3 Central Marmara Rupture System:

The northern strand of the NAFZ forms a major transtensional NW-SE right bend under the Sea of Marmara at the Çınarcık trough (Murru et al., 2016). The fault trace follows the northern margin of the Marmara Sea and connects the complex Central Marmara and Tekirdağ pull-apart basins, before merging into the NE-SW striking Ganos fault on land (Wong et al., 1995; Okay et al., 1999; Armijo et al. 2002; Le Pichon et al., 2001; Yaltirak, 2002; McNeill et al., 2004; Murru et al., 2016). Building the segmentation model for the off-shore segments of NAFZ (also known as the Central Marmara Fault-CMF) is especially difficult because the fault traces are not directly observable (Aksu et al., 2000; Imren et al., 2001; Le Pichon et al., 2001; Armijo et al., 2002, 2005; Pondard et al., 2007). Murru et al. (2016) noted that the segments under Marmara Sea are bounded by geometric fault complexities and discontinuities (e.g., jogs and fault bends) that can act as barriers to rupture propagation (Segall and Pollard, 1980; Barka and Kadinsky-Cade, 1988; Wesnousky, 1988; Lettis et al., 2002; An, 1997) and proposed two separate segments for CMF. We adopted the fault geometry and the segments proposed by Murru et al. (2016) to build the 2-segment Central Marmara rupture system (see Figure 1a for details). As mentioned by Murru et al. (2016), this model is consistent with the segmentation model proposed by Armijo et al. (2002) and in good agreement with the observed Marmara Sea basin morphology and geology (Flerit et al., 2003; Muller and Aydin, 2005; Carton et al., 2007; Pondard et al., 2007; Şengör et al., 2014).

## 2.4 Annual Slip Rates:

Past studies based on GPS measurements (McClusky et al. 2000; Meade et al., 2002; Armijo et al., 2002; Reilinger et al., 2006; Hergert and Heidbach, 2010; Ergintav et al., 2014) suggest a 22 ± 3 mm/yr dextral motion along the major block-bounding structures of the NAFZ, with more than 80% being accommodated along the northern branch.  On this branch, the

5    segments that formed the west and central parts of Izmit rupture system (Segments 3, 2_1, 2_2 and 2_3 in Figure 1a) share the total slip rate with Geyve-Iznik Fault. The slip rate participation among the northern strand of NAFZ and Geyve-Iznik fault was given as 16 mm/yr and 9 mm/yr in Stein et al. (1997). However, Murru et al. (2016) have adopted the annual slip rate of 20±2 mm/yr for the northern strand based on the proposals of Flerit et al. (2003) and Ergintav et al. (2014). Similarly, ~~We~~ we achieved a better fit with the associated seismicity of Izmit rupture system by ~~increasing the share of~~assigning 19±2

10   mm/yr annual slip rate to the northern strand of NAFZ ~~to~~ (please refer to Section 4 for further details). ~~This value is also in good agreement with the annual slip rate given in Murru et al. (2016): they have adopted 20±2 mm/yr based on the proposals of Flerit et al. (2003) and Ergintav et al. (2014).~~ Similarly, the total slip rate is distributed over the eastern segment of NAFZ Southern Strand (Segment 1 in Figure 1a) and the segments of Düzce Rupture System (D1 and D2). Ayhan et al. (2001) suggested that up to 10 mm/yr of the motion is accommodated on the Düzce-Karadere strand of the NAF. We also utilized

the same annual slip rate of 10±2 mm/yr for Düzce_1, Düzce_2 and Karadere segments without any modifications (Table 1).

The mean slip rates adopted for Central and West Marmara sub-segments (19 mm/yr) is consistent with the neighbouring sub-segments of the Izmit and Ganos/Saros rupture systems. Ergintav et al. (2014) noted that the PIF segment (Segment 4) is actively accumulating strain and has not experienced a large event since 1766, making it the most likely segment to generate a M > 7 earthquake. The slip rate estimate given in Ergintav et al. (2014) for the Prince Island Fault and Çınarcık Basin is

15±2 mm/yr. Murru et al. (2016) distributed the annual slip rate of 17 mm/yr among two parallel branches in this zone; 14±2 mm/yr for Çınarcık segment and 3±1 mm/yr for South Çınarcık segment based on the recent works of Ergintav et al. (2014) and Hergert and Heidbach (2010). Therefore, the slip rate value that we have used on the horizontal plane (17 mm/yr) is identical to these recent estimates (Figure 1d). In our analysis, 6±2 mm/yr extension is assigned to the North Çınarcık segment while 3±2 mm/yr is assigned to the South Çınarcık Segment. Since the North Çınarcık Segment was ruptured

during the 17 August 1999 earthquake, we assumed that all the strike-slip motion was taken-up by the North Çınarcık Segment; therefore, all of the 17 mm/yr dextral motion is assigned to the North Çınarcık Segment. The slip rate given for the Central Marmara fault by Ergintav et al. (2014) (2 mm/year) is unusually low compared to the previous estimates and may be suffering from the sparsity of the network and GPS coverage on the north shores of Marmara Sea as mentioned by the authors. For this rupture system, the annual slip rate we adopted (19±2 mm/yr) is in good agreement with the ~~proposal~~

~~of~~value given in Murru et al. (2016) (18±2 mm/yr) and with the seismicity rates based on instrumental earthquake catalogue (Figure 4b).

The slip rate given in the SSC model of Murru et al. (2016) is directly adopted for the Ganos sub-segment whereas; the slip rate partitioned in between the North Saros and South Saros sub-segments in Murru et al. (2016) is concentrated over the

North Saros sub-segment (Table 1). This is because the southern segment is developed in response to transtension exerted by the curvilinear trace of northern segment (Okay et al., 2004), a mechanism somewhat similar to northern and southern Çınarcık segments proposed above. The slip rate assigned to the Ganos and Saros sub-segments is consistent with the recent GPS velocity profiles given in Hergert and ~~Heidbach (2010)~~ ~~et al. (2011)~~ and Ergintav et al. (2014). Table 1 summarizes the references for the utilized annual slip rates for each segment and the uncertainty related to the slip rate included in the logic tree.

## 3 Instrumental Earthquake Catalogue and Activity Rates of Earthquakes

Catalog of earthquakes documenting the available knowledge of past seismicity within the site region is a key component of the seismic source characterization for the hazard analysis. A very detailed review of the historical earthquakes and their rupture zones around the Marmara Sea region was documented by Parsons (2004). These earthquakes and the extension of their rupture zones are directly utilized in this study to define the sub-segments, rupture systems, and to calculate the mean characteristic magnitude values. The Integrated and Homogeneous Turkish Earthquake Catalog published by KOERI (Kalafat et al., 2011) including the events with $M_w > 4$ that occurred between 1900 and 2010 is employed to represent the instrumental seismicity in the region. It is notable that areal source zones (or polygons) are not utilized in the SSC model to estimate the activity rates; therefore, the maximum magnitude estimates and the PSHA results are not solely dependent on the collected catalogue. The mainshock-aftershock classification of the catalog (de-clustering) is performed and the aftershocks are removed from the dataset using the Reasenberg (1985) methodology in the ZMAP software package (Wiemer, 2001) with minimum and maximum look ahead times of 1 and 10 days, and event crack radius of 10 km.

Catalog completeness analysis for different magnitude ranges is performed in order to achieve the catalogue completeness levels used in calculating the magnitude recurrence parameters. Cumulative rates of earthquakes larger than specific magnitude levels are plotted vs. years in order to examine the completeness of catalog as shown in Figure 2. For different cut-off magnitudes, the breaking points for the linear trends in the cumulative rate of events are examined and a significant breaking point is observed to be at 52 years from the end of the catalogue for magnitudes smaller than 4.5 and 5.0. Therefore, the catalog was assumed to be complete for 52 years for $4.0 \leq M_w \leq 4.5$ and $4.5 \leq M_w \leq 5.0$ earthquakes, respectively. Although the larger magnitude plots in Figure 2 suffer from the lack of data due to the truncation of the catalog, the catalog is assumed to be complete for the greater magnitudes for the whole-time span (110 years). The catalogue completeness intervals used in Şeşetyan et al. (2016) and in this study for 4.7<M<5.7 earthquakes are consistent even if the compiled catalogues are different.

The magnitude-frequency relationship developed for each rupture system and the background zone is explained in the next section. Only one of the magnitude-frequency relationship parameters, the slope of the cumulative rate of events (as known as the b-value), is calculated based on the compiled catalogue. We delineated three different zones for estimating the b-value

considering the temporal and spatial variability of this parameter as shown in Figure 1c. Zone 1 includes the Ganos/Saros and Central Marmara rupture systems, Zone 2 covers the Izmit and Düzce rupture systems, and Zone 3 is a larger area that includes both Zone 1 and 2. For each zone, the b-value is estimated using the maximum likelihood method provided in ZMAP software package. Figure 3 (a-c) shows the completeness magnitudes and the b-values for Zones 1, 2, and 3. Analysis results show that the b-value varies in between 0.68 and 0.74 for different rupture systems given in the previous section; whereas, the b-value for the large area covering whole system is equal to 0.76.

Additionally, the b-values for each zone are estimated using the modified maximum likelihood method (Weichert, 1980) that takes into account the completeness of the catalog for different magnitude bins. The b-values calculated by Weichert (1980) method is approximately 5% higher than the maximum likelihood estimations of ZMAP for Zones 1 and 2, but for the larger zone (Zone 3), estimated b-values are almost the same in both methods (Table 2). To acknowledge the uncertainty in the b-value estimations, 30% weight is assigned to the zone-specific b-value calculated by ZMAP and the zone-specific b-value calculated using Weichert (1980) method each, and 40% weight is given to the regional b-value since the number of data in this zone is larger and the estimated b-value is statistically more stable. Finally, the b-value for the background zone (limits shown in Figure 5) is calculated as 0.81 by removing the earthquakes within the buffer zones. Uncertainty in the b-value of background zone is determined using the method proposed by Shi and Bolt (1982) and included in the logic tree (Table 2).

Estimated b-values are relatively small when compared to the b-values estimated for large ares (b≈1); however, our findings are consistent with the current literature. Şeşetyan et al. (2016) provided a thorough analysis of the b-value for the whole Turkish territory and proposed that b=0.77 for Central Marmara region and b=0.67 for North Anatolian Fault Zone (Figure 15 of Şeşetyan et al., 2016). The small differences in the b-values proposed by Şeşetyan et al. (2016) and the b-values estimated in this study are due to the geometry of the selected zones and the differences in the compiled catalogues. The b-value used by Moschetti et al. (2015) for Western United States (b=0.8) is not very different than our estimates.

## 4 Magnitude Recurrence Models – Seismic Moments

Seismic sources can generate varied sizes of earthquakes and magnitude distribution models describe the relative rate of these small, moderate and large earthquakes. The basic and the most common magnitude frequency distribution (MFD) is the exponential model proposed by Gutenberg and Richter (1944) (G-R). Since there is a maximum magnitude that the source can produce and a minimum magnitude for engineering interest, the G-R distribution is usually truncated at both ends and renormalized so that it integrates to unity. The truncated exponential MFD (Cosentino et al., 1977) is given in Eq. (1):

$$f_m^{TE}(M) = \frac{\beta \exp\left(-\beta(M - M_{min})\right)}{1 - \exp\left(-\beta(M_{max} - M_{min})\right)} \tag{1}$$

where $\beta = ln(10) \times b - value$, $M_{min}$ is the minimum magnitude, and $M_{max}$ is the maximum magnitude. Youngs and Coppersmith (1985) proposed that the truncated exponential distribution is suitable for large regions or regions with multiple

faults but in most cases does not work well for individual faults. Instead, individual faults may tend to rupture at what have been termed as *"characteristic"* size events and the alternative magnitude distribution for this case is the characteristic model proposed by Schwartz and Coppersmith (1984). In characteristic MFD, once a fault begins to rupture in large earthquakes, it tends to rupture the entire fault segment and produce similar size earthquakes due to the geometry of the fault.

It is notable that the characteristic model does not consider the small-to-moderate magnitude earthquakes on a fault. A third model was proposed by Youngs and Coppersmith in 1985 that combines the truncated exponential and characteristic magnitude distributions as shown in Eq. (2) and (3):

$$
f_m^{YC}(M) = \begin{cases} \dfrac{1}{1+c_2} \times \dfrac{\beta \exp\left(-\beta(\bar{M}_{char}-M_{min}-1.25)\right)}{1-\exp\left(-\beta(\bar{M}_{char}-M_{min}-0.25)\right)} & \text{for } \bar{M}_{char}\text{-}0.25<M\leq\bar{M}_{char}\text{+}0.25 \\[4mm] \dfrac{1}{1+c_2} \times \dfrac{\beta \exp\left(-\beta(M-M_{min})\right)}{1-\exp\left(-\beta(\bar{M}_{char}-M_{min}-0.25)\right)} & \text{for } M_{min}<M\leq\bar{M}_{char}\text{-}0.25 \end{cases}
\tag{2}
$$

where,

$$
c_2 = \frac{0.5\beta \exp(-\beta(\bar{M}_{char} - M_{min} - 1.25))}{1 - \exp(-\beta(M_{char} - M_{min} - 0.25))}
\tag{3}
$$

and $M_{char}$ is the characteristic earthquake magnitude. Coupling the truncated exponential MFD with seismic sources defined by planar fault geometries results in unrealistically high rates for small-to-moderate magnitude events (Hecker et al., 2013),

especially in the close vicinity of NAFZ (Gülerce and Vakilinezhad, 2015). Therefore, the composite MFD proposed by Youngs and Coppersmith (1985) is utilized to represent the relative rates of small, moderate and large magnitude earthquakes generated by rupture sources defined in this study.

The rupture systems presented in Section 2 includes more than one sub-segment. We adopted the terminology of WGCEP

(2003) and defined the rupture source as a fault sub-segment or a combination of multiple adjacent fault sub-segments that may rupture and produce an earthquake in the future. For Düzce, Central Marmara, and Ganos/Saros rupture systems with two sub-segments (as A and B), three different rupture sources can be defined; single segment sources (A and B) and a two-sub-segment source (A+B). Any possible combination of rupture sources that describes the complete rupture of the system is defined as the rupture scenario. Two rupture scenarios for these rupture systems are; (1) rupture of the two sub-segments

individually and (2) rupture of the two sub-segments together. The rupture model includes the weighted combination of rupture scenarios of the rupture system. Five segments defined for Izmit rupture systems form a rupture model with 15 rupture sources and 16 rupture scenarios (Table 5). The minimum magnitude ($M_{min}$) is set to $M_w$=4.0 for all rupture sources considering the completeness magnitude. Mean characteristic magnitudes ($M_{char}$) for each rupture source are calculated using the relationships proposed by Wells and Coppersmith (1994) and Hanks and Bakun (2014). The $M_{char}$ values calculated using

both equations are quite close to each other and the absolute value of the difference is smaller than 0.13 in magnitude units (Table 6). To grasp the epistemic uncertainty, average of the $M_{char}$ value from both methods are utilized in the center of the logic tree with 50% weight and both the $M_{char}$ -0.15 and $M_{char}$ +0.15 values are included by assigning 25% weight. The upper

bound for the magnitude PDF ($M_{max}$) is determined by adding 0.25 and 0.5 magnitude units to $M_{char}$ for each source in each logic tree branch (Table 6).

MFD only represents the relative rate of different magnitude earthquakes. In order to calculate the absolute rate of events, the activity rate $N(M_{min})$ defined as the rate of earthquakes above the minimum magnitude should be used. For areal sources, $N(M_{min})$ may be calculated by using the seismicity within the defined area. For planar fault sources, the activity rate is defined by the balance between the accumulated and released seismic moments as shown in Eq. (4). The accumulated seismic moment is a function of the annual slip rate (S) in cm/years, area of the fault (A in $cm^2$) and the shear modulus of the crust ($\mu = 30 \times 10^{12}$ dyne/$cm^2$, Brodsky et al., 2000; Field et al., 2009). The S for the rupture sources that includes more than one segment with different S values are calculated using the weighted average of annual slip rates (weighs are determined based on the area of the segment as shown in Eq. 5).

$$N(M_{min}) = \frac{\mu A S}{\int_{M_{min}}^{M_{max}} f_m(M_w) 10^{1.5 M_w + 16.05} dM} \tag{4}$$

$$S_{source} = \frac{\sum_{all\ segments\ for\ the\ source} S_{segment} \times A_{segment}}{\sum_{all\ segments\ for\ the\ source} A_{segment}} \tag{5}$$

Ultimately the MFD and the activity rate are used to calculate the magnitude recurrence relation, $N(M)$, as shown in Eq. (6).

$$N(M) = N(M_{min}) \int_{M_{min}}^{M_{max}} f_m(M_w)\, dM \tag{6}$$

The magnitude recurrence relation given in Eq. (6) and the accuracy of the model parameters such as the b-value or $M_{max}$ shall be tested by the relative frequency of the seismicity associated with the source in the moment-balanced PSHA procedure. Therefore, a weight is assigned to each rupture scenario and the cumulative rates of events attributed to that particular rupture system are plotted along with the weighted average of the rupture scenarios to calibrate the assigned weights and to evaluate the balance of the accumulated and released seismic moment. The "*moment-balancing*" graphs for Izmit, Düzce, Central Marmara, and Ganos/Saros rupture systems are provided in Figure 4 and used to compare the modelled seismicity rate with the instrumental earthquake catalogue. In these plots, the black dots stand for the cumulative annual rates of earthquakes and the error bars represent the uncertainty introduced by unequal periods of observation for different magnitudes (Weichert, 1980). In Figure 4, the scenarios that are separated by plus signs in the legend are the scenarios with multiple rupture sources. When multiple segments rupture together, these scenarios are separated by a comma sign in the legend. For example, the "S4, S5" line in Figure 4(c) represents the scenario where S4 and S5 sub-segments are ruptured individually. This scenario brings in relatively higher rates for small-to-moderate earthquakes when compared to the S4+S5 scenario which represents the rupture of these two segments together to produce a larger event.

The best fit between the cumulative annual rate of events and the weighted average of rupture scenarios (red dashed lines) is established by modifying the weights of the rupture scenarios by visual interpretation. To achieve a good fit, the seismic

source modeller needs to understand the contribution of the magnitude recurrence model parameters to the red broken line in different magnitude ranges. For example, the b-value significantly affects the small magnitude portion of the curve since the Youngs and Coppersmith (1985) magnitude PDF is ~~utilized~~used. Please remind that the b-value is calculated based on the same catalogue but for a larger region. Defining a large number of sub-segments for a rupture system also increases the cumulative rate of small magnitude events. The good fit in the small magnitude range of Figure 4 shows that: i) the b-value calculated using the larger zone is compatible with the seismicity associated with the planar source, ii) utilized segmentation model is consistent with the relative rates of small-to-moderate and large events, and iii) annual slip rate is compatible with the seismicity over the fault. The large magnitude rates in Figure 4 are poorly constrained since the catalogue used herein only covers 110 years and that time span is obviously shorter than the recurrence rate for the large magnitude event. Hecker et al. (2013) explained that by "*rates of large-magnitude earthquakes on individual faults are so low that the historical record is not long enough to test this part of the distribution*" and suggested using the "*inter-event variability of surface-rupturing displacement at a point as derived from geologic data sets*" to test the upper part of the earthquake-magnitude distribution. In each moment balancing plot, relatively higher weights are assigned to the rupture scenarios that combine the individual (single-segment) rupture sources based on the assumption (and modeller's preference) that single-segment ruptures are more likely than multiple-segment ruptures. The weights assigned to each rupture scenario are given in Table 4.

## 5 Background Zone – Smoothed Seismicity

A background source zone of diffused seismicity is utilized to characterize the seismicity that is not associated with the rupture systems described in the previous sections. This additional background source zone represents the seismicity associated with the mapped active faults on the south of Marmara Sea (orange fault lines in Figure 1a) and the interpretation that even in areas where active faults or distinctive zones of seismicity clusters are not observed, earthquakes can still occur. Figure 1c shows that the spatial distribution of the earthquakes (outside the buffer zones around the rupture systems) is not homogeneous; density of the events increases significantly around the Geyve-Iznik Fault Zone. Therefore, defining an areal source zone with homogeneous seismicity distribution would result in the overestimation of the seismic hazard in Istanbul. Instead, the background source is modelled as a source of gridded seismicity where the earthquakes are represented as point or planar fault sources at the ~~centers~~centres of evenly spaced grid cells (0.05 degree spacing). The truncated exponential magnitude distribution (Eq. 1) is selected to represent the relative frequency of the different magnitude events for this source. In the magnitude recurrence model, spatially uniform $M_{max}$ and b-values and spatially variable a-values, or seismicity rates, are defined. The minimum magnitude ($M_{min}$) is again set to $M_w$=4.0 and the b-value is taken as 0.81. The a-value for each grid cell was calculated from the maximum likelihood method of Weichert (1980), based on events with magnitudes of 4.0 and larger. The gridded a-values were then smoothed by using an isotropic Gaussian kernel with a correlation distance of 10 km (Frankel, 1995). The smoothed-seismicity rates overlying the earthquakes outside the buffer zones are presented in Figure 5. Tabulated values of the grid cell coordinates and the seismicity rates are provided in the Electronic Supplement 2.

During the calculations of the smoothed seismicity rates, the earthquakes in buffer zones are not included in smoothing (and not double-counted). The buffer zones are only used to "associate" the earthquakes with the fault zones and collapse the earthquakes to the vertical fault planes. Therefore, the background source and the fault sources can be superposed in the PSHA calculations.

The $M_{max}$ distribution of the background zone is developed by taking into account the lack of evidence for surface faulting in the city of Istanbul. So far, no active fault has been reported from the near vicinity of the study area. Similarly, the MTA Active Fault Maps (Emre et al., 2013) do not contain any active fault in the northern part of the NAFZ between Izmit and Tekirdağ. Moschetti et al. (2015) mentioned that the development of the $M_{max}$ model for shallow crustal seismicity in the Western United States benefits from the large set of regional earthquake magnitudes from the historical and paleoseismic

records; however, the background seismicity model accounts for earthquake ruptures on unknown faults; therefore, the $M_{max}$ distribution must reflect the range of possible magnitudes for these earthquakes. We adopted a similar approach based on the fault segments of the southern strand of NAFZ documented in Murru et al. (2016) and calculated the characteristic magnitude for each segment using Wells and Coppersmith (1994) magnitude-rupture area relation. Based on the estimations of characteristic magnitude of earthquakes that may occur on the southern strand of NAFZ, the logic tree for $M_{max}$ (centered

on $M_w = 6.8$) of the background zone is developed (Table 6). The focal mechanisms of the background source should reflect the tectonic style of the parent region; therefore, a weighted combination of strike-slip (SS, 75%), normal (N, 20%), and reverse (R, 5%), motion with weights that sum to 1 is assigned to this source (Table 3). A uniform distribution of focal depths between the surface and 18 km depth is utilized (Emre et al., 2016).

## 6 Discussions on the Uncertainty Involved in the Proposed SSC Model

In the proposed SSC model, the uncertainties related to $M_{max}$, magnitude-rupture area relations, magnitude recurrence model parameters, and the annual slip rates are considered and included in the logic tree (Supplement #2). On the other hand, the uncertainty related to the fault geometry such as the uncertainty in segment lengths, fault widths, and dip angles remained unexplored. All rupture sources within each rupture system are considered to occur in order to capture the aleatory variability in the extent and potential size of future ruptures. However, the epistemic uncertainty in the potential rupture

scenarios are not ~~considered~~ taken into account since only one set of weights for each rupture scenario is included in the logic tree. In order to compare the epistemic uncertainty in the proposed SSC model with the uncertainty in the earthquake catalogue, the SSC model fractals for each rupture source are calculated and the extreme values represented by the single-segment rupture sources and full-span rupture source are presented in Figure 6 by red and blue sets of curves, respectively. It is notable that the rate of observed earthquakes were used to validate the rupture scenario weights in Figure 4, aiming to

capture a good fit between weighted average rates and the mean rates of observed earthquakes. Figure 6 shows that the uncertainty range sampled by the proposed model is consistent with the rate of earthquakes associated with each rupture system, especially for $M_w < 6$ events that have large sample size.

We would like to emphasise that the SSC model presented here is different than the models proposed by Gülerce and Ocak (2013) and Murru et al. (2016): differences in the fault geometry is minor but the differences in the magnitude recurrence models and the time dependent probabilities of earthquakes are more significant. Unfortunately, earlier publications did not provide enough information on earthquake rates for doing case-to-case comparison of the earthquake rates proposed herein with the previous works. Our model does not utilize the time-dependent hazard methodologies as in Murru et al. (2016); however, we believe that the ongoing research on the paleoseismic recurrence periods (National Earthquake Strategy and Action Plan for 2023, NESAP-2023) will provide a substantial contribution in the PSHA practice of Turkey and eventually will lead to a change the hazard estimates. The available paleoseismic data on NAFZ are few and insufficient to provide meaningful constraints on the "grand inversion" as used in UCERF3 model for California (Field et al., 2014). Therefore, proposed model does not include fault-to-fault ruptures that can jump over the boundaries of the defined rupture systems.

**7 Conclusions**

This manuscript presents the details of the SSC model proposed for the PSHA studies in Istanbul. When compared to the previous SSC models developed for this region, significant improvements in the proposed model can be listed as follows: (1) planar seismic sources that accounts for the most current tectonic information (e.g. updated fault maps) are built, (2) the multi-segment rupture scenarios are systematically utilized in the rupture forecast, (3) buffer zones around the rupture systems are defined to associate the small, moderate, and large magnitude events with the rupture systems, (4) activity rates for the planar rupture systems are calculated using the geological and geodedic constraints (e.g. slip rate and fault geometry), (5) balance of the accumulated and released seismic moment is considered in building the magnitude recurrence model, and (6) associated earthquakes are used to test the suitability of the magnitude recurrence model with the instrumental seismicity rates. Even though the rupture systems develop in this study accounts for the relative rates of small, moderate, and large magnitude events that can occur on the faults, a background source is defined to represent the small-to-moderate magnitude earthquakes that may take place anywhere in the vicinity of Istanbul and Marmara Sea. Properties of the rupture systems and background source, the logic tree associated with both of these components, coordinate of the fault segments, and smoothed seismicity rates are fully documented throughout the text and in the Electronic Supplements 1-2. Therefore, proposed SSC model can be directly implemented to any of the available PSHA software for the site-specific PSHA analysis in Istanbul. We would like to underline that the geometry and the earthquake rates of the background source may be modified for any application outside the greater Istanbul area. The hazard analyst can incorporate the full rupture model and the complete logic tree provided in this manuscript to most of the available hazard codes without explicitly calculating the earthquake rates. In case that the earthquake rate has to be incorporated to the hazard code; the earthquake rates for each branch of the logic tree given in Electronic Supplement#3 can be used.

**Acknowledgements**

Authors of this manuscript are grateful for the support provided by Turkish Atomic Energy Authority (TAEK). This work was partially supported by the Pacific Gas & Electric Company Geosciences Department. The authors are thankful to the guest editors and anonymous reviewers for their insightful comments.

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

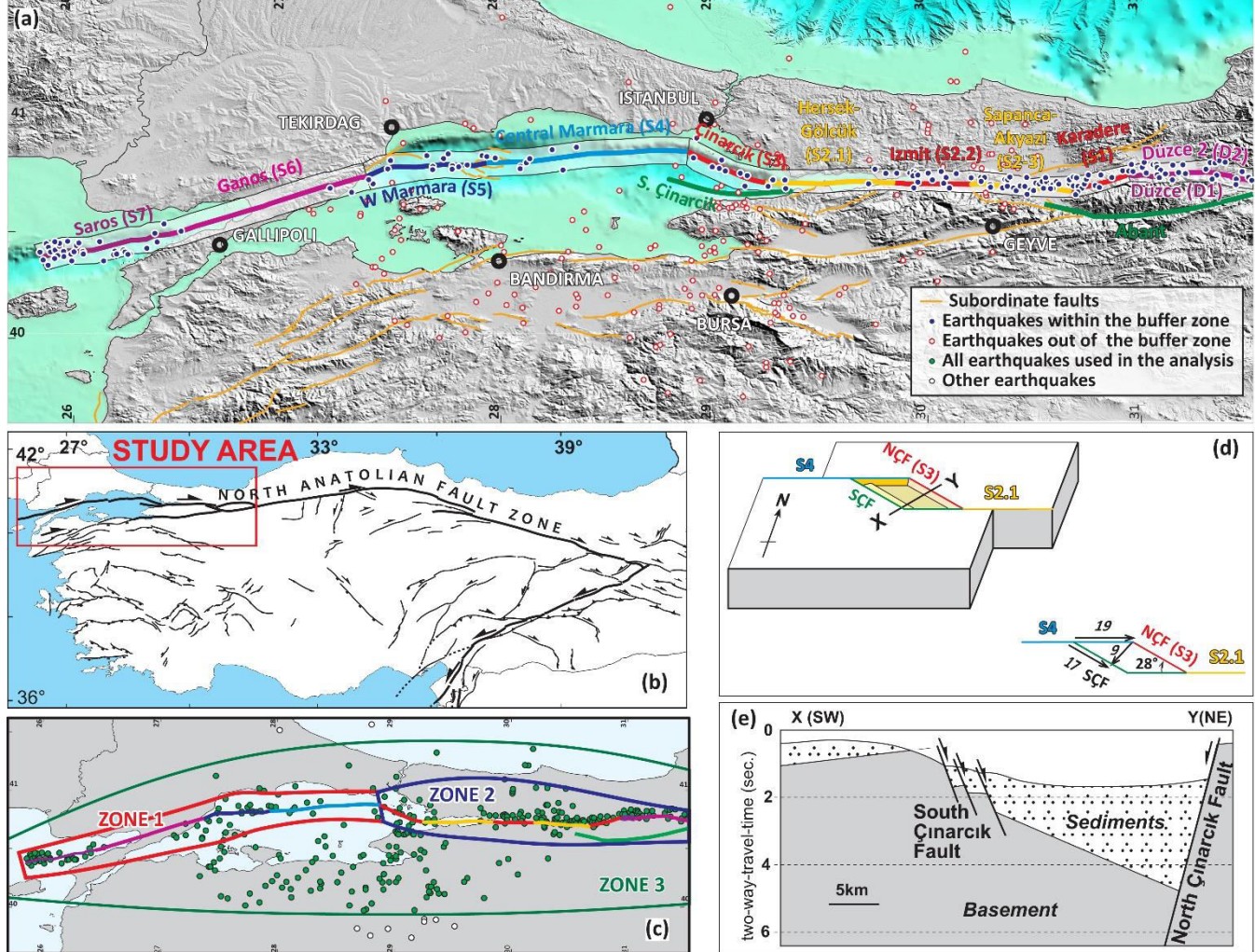

**Figure 1: (a)** Major branches of North Anatolian Fault Zone, defined rupture systems and the instrumental seismicity ($M_w$>4) in the study area. The buffer zones used for source-to-epicenter matching are shown around the rupture systems. **(b)** Simplified active tectonic scheme of Turkey (modified from Emre et al., 2013). Thick lines are North Anatolian and East Anatolian fault zones, thin lines are other active faults. **(c)** Distribution of the declustered seismicity used to calculate the b-values. Zone 1, Zone 2 and Zone 3 are the polygons used to calculate the b-values. **d)** Slip distribution model for Çınarcık segment. Right bending of North Çınarcık segment is 28° with respect to Central Marmara and Hersek-Gölcük segments. This results in 17 mm/y slip along the North Çınarcık segment (NÇF) and 9mm/y normal slip transverse to the fault. This 9mm/y slip is the total slip on North and South Çınarcık faults (SÇF). **e)** Simplified geometries of Çınarcık faults delimiting the Çınarcık Basin based on seismic profile of Laigle et al. (2008) almost passing through the line XY. ~~Note that the throw of North Çınarcık Fault is almost twice that of South Çınarcık Fault.~~

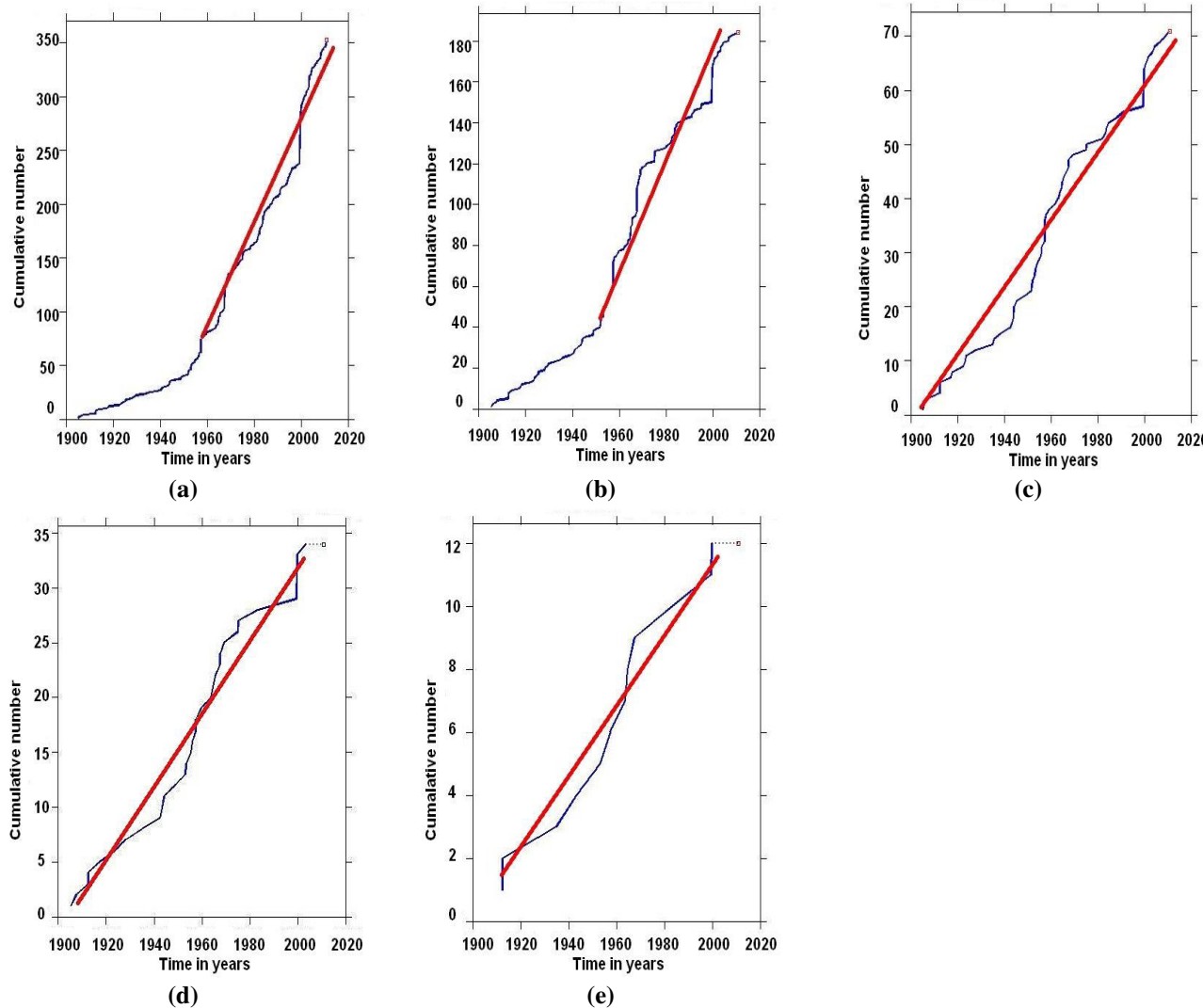

**Figure 2: The catalogue completeness analysis for the instrumental earthquake catalogue showing the cumulative number of events for (a) Mw ≥4.0, (b) Mw ≥4.5, (c) Mw ≥5.0, (d) Mw ≥5.5, and (e) Mw ≥6.0.**

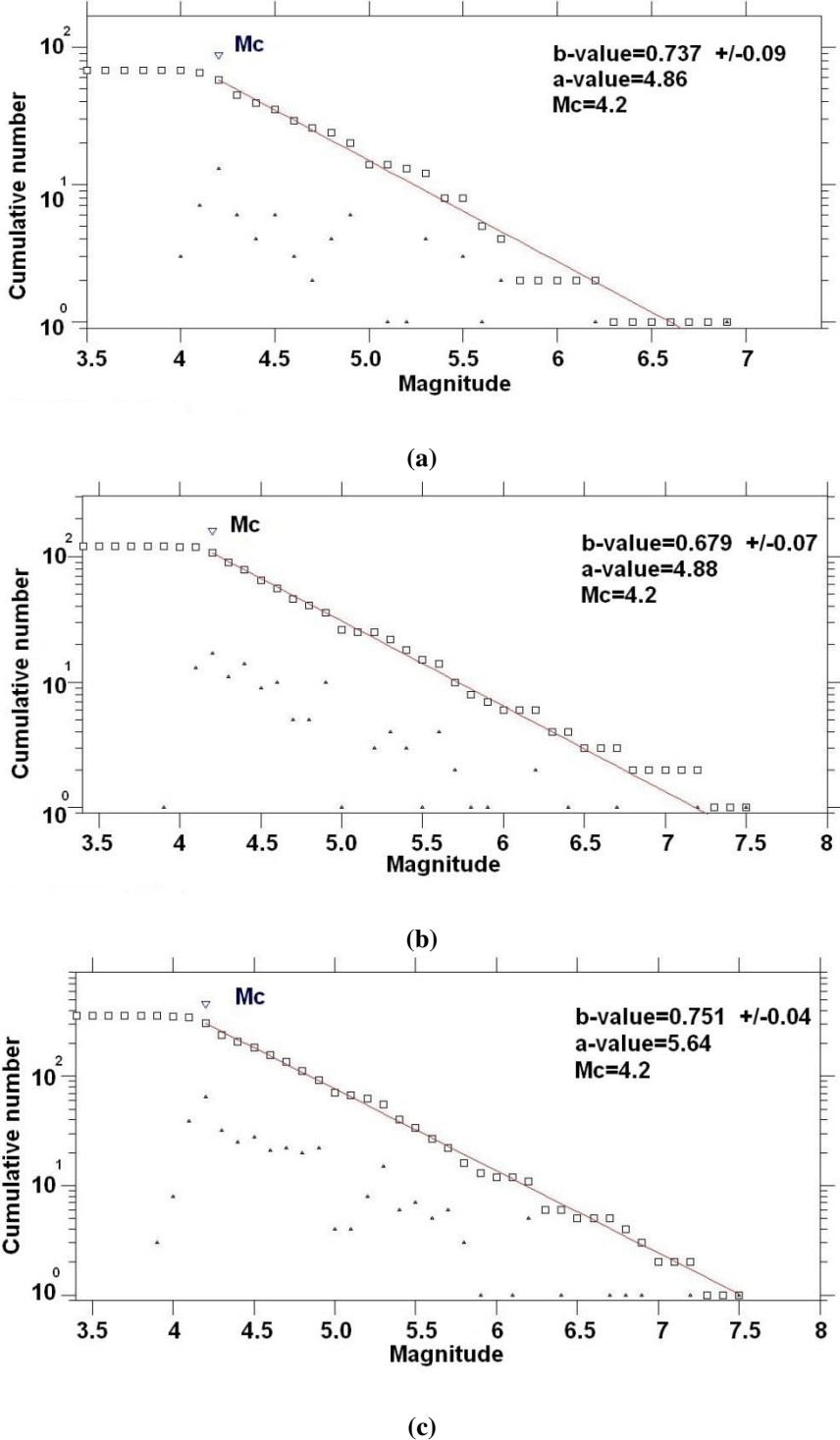

**Figure 3: Estimated magnitude recurrence parameters for (a) Zone 1, (b) Zone 2, and (c) Zone 3.**

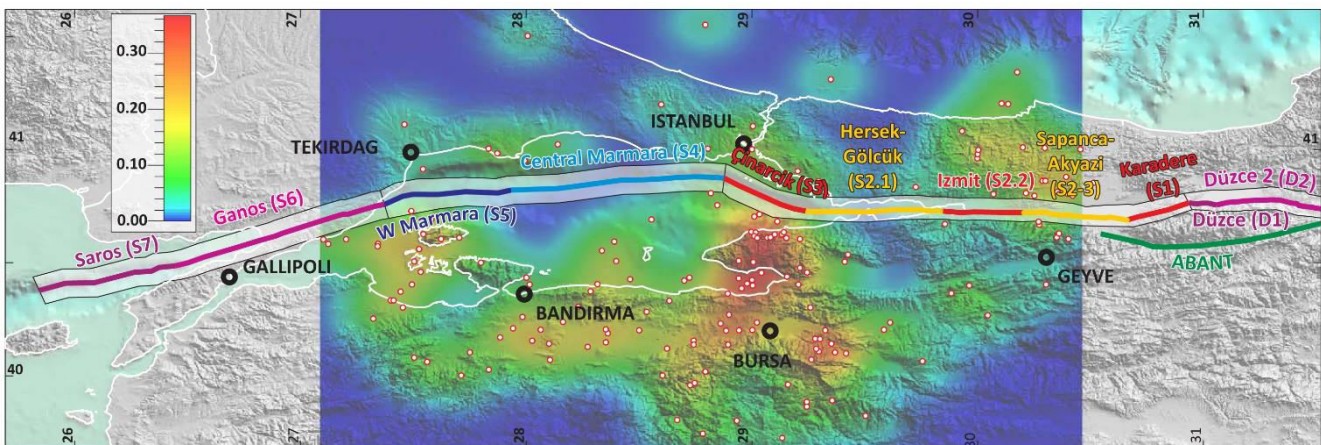

**Figure 4: Cumulative rates of earthquakes for the magnitude recurrence model and associated events (moment balancing graphs) for (a) Izmit, (b) Düzce, (c) Central Marmara, and (d) Ganos/Saros rupture systems. Black points are the earthquakes associated with the rupture system, purple and blue lines show the single-segment and multi-segment ruptures, red broken line is the weighted average of the magnitude recurrence model. In these graphs, the median values of the slip rates and M$_{max}$ and zone-specific b-values are utilized.**

**Figure 5: Spatial distribution of the activity rates in the smoothed seismicity source. Red circles are the earthquakes used in the analysis.**

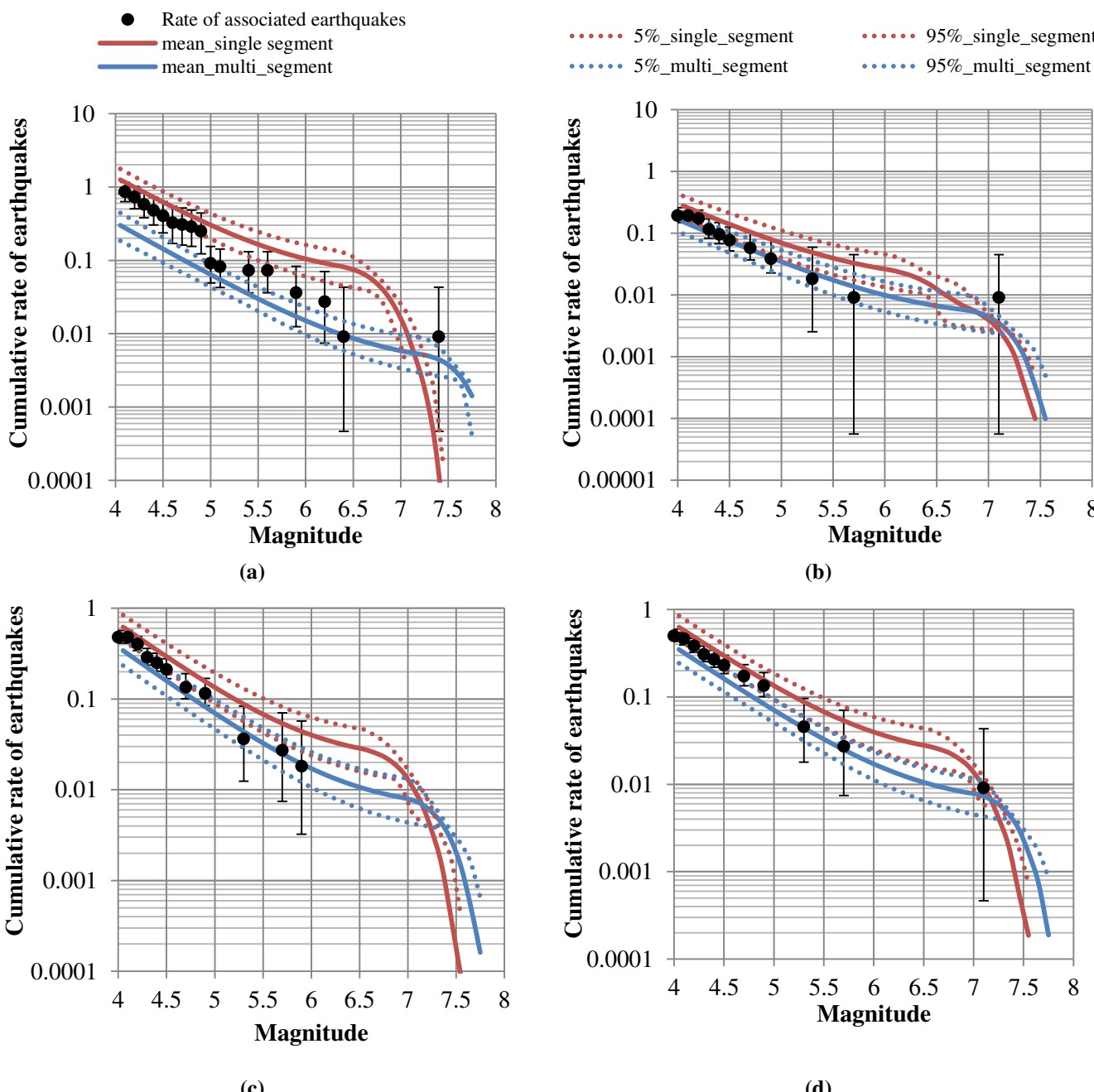

**Figure 6: Mean and fractals of the single-segment and multi-segment rupture scenarios with the cumulative rate of earthquakes associated with the rupture system for (a) Izmit, (b) Düzce, (c) Central Marmara, and (d) Ganos/Saros rupture systems. Solid lines are the mean rates and the dashed lines show the 5% and 95% rates for each rupture scenario.**

**Table 1: The fault segments and rupture systems included in the SSC model. References given in the last column are: 1) Flerit et al. (2004), 2) Murru et al. (2016), 3) Ergintav et al. (2014), 4) Ayhan et al. (2001), 5) Hergert et al. (2011). Weights associated with the mean, upper bound and lower bound are 0.5, 0.25, and 0.25, respectively.**

| Rupture System | Segment No | Segment Name | Length (km) | Width (km) | Slip Rate and associated uncertainty (mm/yr) | Reference for the slip rate estimation |
|---|---|---|---|---|---|---|
| Izmit | 3 | North Çınarcık | 34.6 | 18 | 17±2 (6±2 extension) | 1, 2, 3, Figure 1 |
| Izmit | 2_1 | Hersek- Gölcük | 51.6 | 18 | 19±2 | 1, 2, 3 |
| Izmit | 2_2 | İzmit | 30.2 | 18 | 19±2 | 1, 2, 3 |
| Izmit | 2_3 | Sapanca – Akyazı | 39.1 | 18 | 19±2 | 1, 2, 3 |
| Izmit | 1 | Karadere | 24.7 | 18 | 10±2 | 1, 4 |
| Düzce | D1 | Düzce_1 | 10.5 | 25 | 10±2 | 1, 4 |
| Düzce | D2 | Düzce_2 | 41 | 25 | 10±2 | 1, 4 |
| Ganos/Saros | 6 | Ganos | 84 | 15 | 19±1 | 1, 3, 4, 5 |
| Ganos/Saros | 7 | Saros | 53 | 15 | 19±1 | 1, 3, 4, 5 |
| Central | 4 | Central Marmara | 80 | 15 | 19±2 | 1, 2 |
| Central | 5 | West Marmara | 49 | 15 | 19±2 | 1, 2 |
| Çınarcık | 8 | South Çınarcık | 39 | 18 | (3±2 extension) | 2, Figure 1 |

5   **Table 2: b-values estimated using different methods and corresponding weights in the logic tree.**

| Source Zone | Maximum likelihood estimation by ZMAP (Zone-specific) | | Maximum likelihood estimation by Weichert (1980) (Zone-specific) | | Regional Value | |
|---|---|---|---|---|---|---|
| | **b-value** | **weight** | **b-value** | **weight** | **b-value** | **weight** |
| Düzce Rupture System | 0.68 | 0.3 | 0.72 | 0.3 | 0.76 | 0.4 |
| Izmit Rupture System | 0.68 | 0.3 | 0.72 | 0.3 | 0.76 | 0.4 |
| Central Marmara Rupture System | 0.74 | 0.3 | 0.78 | 0.3 | 0.76 | 0.4 |
| Ganos/Saros Rupture System | 0.74 | 0.3 | 0.78 | 0.3 | 0.76 | 0.4 |
| | Maximum likelihood estimation by Weichert (1980) (Mean - 2σ) | | Maximum likelihood estimation by Weichert (1980) (Mean) | | Maximum likelihood estimation by Weichert (1980) (Mean + 2σ) | |
| Background Zone | **b-value** | **weight** | **b-value** | **weight** | **b-value** | **weight** |
| | 0.714 | 0.20 | 0.81 | 0.60 | 0.906 | 0.20 |

**Table 3: Aleatory variability for style of faulting in the background zone**

| | Style of faulting | | | |
|---|---|---|---|---|
| Weights | Normal | Strike Slip | Reverse | Normal-oblique |
| 150km radius background zone | 0.20 | 0.75 | 0.05 | |
| All segments except Çınarcık Fault | - | 1.00 | - | |
| North and South Çınarcık Segments | - | - | - | 1.0 |

**Table 4: Aleatory variability  in the rupture scenario weights**

| Rupture System | Rupture type | Included sub-segment no | Weight |
|---|---|---|---|
| Düzce Rupture System | Single segment ruptures | D1, D2 | 0.5 |
| | 2-segment ruptures | D1+D2 | 0.5 |
| Central Marmara Rupture System | Single segment ruptures | 4,5 | 0.6 |
| | 2-segment ruptures | 4+5 | 0.4 |
| Ganos/Saros Rupture System | Single segment ruptures | 6,7 | 0.6 |
| | 2-segment ruptures | 6+7 | 0.4 |
| Izmit Rupture System | Table 5 | | |

**Table 5: Rupture sources and rupture scenarios utilized for the Izmit rupture system***

| | 3 | 2_1 | 2_2 | 2_3 | 1 | 3+2_1 | 2_1+2_2 | 2_2+2_3 | 2_3+1 | 3+2_1+2_2 | 2_1+2_2+2_3 | 2_2+2_3+2_4 | 3+2_1+2_2+2_3 | 2_1+2_2+2_3+1 | 3+2_1+2_2+2_3+1 | Rupture Scenario | Weight |
|---|---|---|---|---|---|---|---|---|---|---|---|---|---|---|---|---|---|
| 3, 2_1, 2_2, 2_3, 1 | 1 | 1 | 1 | 1 | 1 | 0 | 0 | 0 | 0 | 0 | 0 | 0 | 0 | 0 | 0 | 1 | 0.20 |
| 3+2_1 ,2_2, 2_3, 1 | 0 | 0 | 1 | 1 | 1 | 1 | 0 | 0 | 0 | 0 | 0 | 0 | 0 | 0 | 0 | 2 | 0.07 |
| 3, 2_1+2_2, 2_3 ,1 | 1 | 0 | 0 | 1 | 1 | 0 | 1 | 0 | 0 | 0 | 0 | 0 | 0 | 0 | 0 | 3 | 0.07 |
| 3, 2_1, 2_2+2_3, 1 | 1 | 1 | 0 | 0 | 1 | 0 | 0 | 1 | 0 | 0 | 0 | 0 | 0 | 0 | 0 | 4 | 0.07 |
| 3, 2_1, 2_2, 2_3+1 | 1 | 1 | 1 | 0 | 0 | 0 | 0 | 0 | 1 | 0 | 0 | 0 | 0 | 0 | 0 | 5 | 0.07 |
| 3+2_1+2_2, 2_3, 1 | 0 | 0 | 0 | 1 | 1 | 0 | 0 | 0 | 0 | 1 | 0 | 0 | 0 | 0 | 0 | 6 | 0.05 |
| 3, 2_1+2_2+2_3, 1 | 1 | 0 | 0 | 0 | 1 | 0 | 0 | 0 | 0 | 0 | 1 | 0 | 0 | 0 | 0 | 7 | 0.05 |
| 3, 2_1, 2_2+2_3+1 | 1 | 1 | 0 | 0 | 0 | 0 | 0 | 0 | 0 | 0 | 0 | 1 | 0 | 0 | 0 | 8 | 0.05 |
| 3+2_1+2_2+2_3, 1 | 0 | 0 | 0 | 0 | 1 | 0 | 0 | 0 | 0 | 0 | 0 | 0 | 1 | 0 | 0 | 9 | 0.05 |
| 3, 2_1+2_2+2_3+1 | 1 | 0 | 0 | 0 | 0 | 0 | 0 | 0 | 0 | 0 | 0 | 0 | 0 | 1 | 0 | 10 | 0.03 |
| 3+2_1, 2_2+2_3, 1 | 0 | 0 | 0 | 0 | 1 | 1 | 0 | 1 | 0 | 0 | 0 | 0 | 0 | 0 | 0 | 11 | 0.03 |
| 3, 2_1+2_2, 2_3+1 | 1 | 0 | 0 | 0 | 0 | 0 | 1 | 0 | 1 | 0 | 0 | 0 | 0 | 0 | 0 | 12 | 0.03 |
| 3+2_1+2_2, 2_3+1 | 0 | 0 | 0 | 0 | 0 | 0 | 0 | 0 | 1 | 1 | 0 | 0 | 0 | 0 | 0 | 13 | 0.03 |
| 3+2_1, 2_2+2_3+1 | 0 | 0 | 0 | 0 | 0 | 1 | 0 | 0 | 0 | 0 | 0 | 1 | 0 | 0 | 0 | 14 | 0.03 |
| 3+2_1, 2_2, 2_3+1 | 0 | 0 | 1 | 0 | 0 | 1 | 0 | 0 | 1 | 0 | 0 | 0 | 0 | 0 | 0 | 15 | 0.03 |
| 3+2_1+2_2+2_3+1 | 0 | 0 | 0 | 0 | 0 | 0 | 0 | 0 | 0 | 0 | 0 | 0 | 0 | 0 | 1 | 16 | 0.14 |
| Rupture Source No | 1 | 2 | 3 | 4 | 5 | 6 | 7 | 8 | 9 | 10 | 11 | 12 | 13 | 14 | 15 | | |

*Note: Rows show the rupture scenarios and the columns show the rupture sources. 1 and 0 in a cell indicate that the particular rupture source is included or excluded in the rupture scenario, respectively. Scenario weights are given in the last column. For sub-segments 3, 2_1, 2_2, 2_3, and 1, please refer to Figure 1b.

**Table 6: Logic tree representing epistemic uncertainty in maximum magnitudes. Weights for Mmax_1, Mmax_2, and Mmax_3 are 0.25, 0.5, and 0.25, respectively. (WC94: Wells and Coppersmith (1994) and HB14: Hanks and Bakun (2014) magnitude-rupture area relation)**

| Rupture System | Rupture Source | Source Width (km) | Source Length (km) | Characteristic Magnitude (WC94) | Characteristic Magnitude (HB14) | $M_{max}$ 1 | $M_{max}$ 2 | $M_{max}$ 3 |
|---|---|---|---|---|---|---|---|---|
| Düzce | D1 | 25 | 10.5 | 6.45 | 6.40 | 6.52 | 6.67 | 6.82 |
| Düzce | D2 | 25 | 41 | 7.05 | 7.06 | 7.16 | 7.31 | 7.46 |
| Düzce | D1+D2 | 25 | 51.5 | 7.15 | 7.19 | 7.27 | 7.42 | 7.57 |
| Central Marmara | S4 | 15 | 80 | 7.12 | 7.15 | 7.23 | 7.38 | 7.53 |
| Central Marmara | S5 | 15 | 49.2 | 6.91 | 6.89 | 7.00 | 7.15 | 7.30 |
| Central Marmara | S4+S5 | 15 | 129.2 | 7.33 | 7.41 | 7.47 | 7.62 | 7.77 |
| Ganos / Saros | S6 | 15 | 84 | 7.14 | 7.18 | 7.26 | 7.41 | 7.56 |
| Ganos / Saros | S7 | 15 | 53 | 6.94 | 6.93 | 7.03 | 7.18 | 7.33 |
| Ganos / Saros | S6+S7 | 15 | 137 | 7.36 | 7.44 | 7.50 | 7.65 | 7.80 |
| Izmit | 3 | 18 | 34.6 | 6.83 | 6.79 | 6.91 | 7.06 | 7.21 |
| Izmit | 2_1 | 18 | 51.6 | 7.01 | 7.01 | 7.11 | 7.26 | 7.41 |
| Izmit | 2_2 | 18 | 30.2 | 6.77 | 6.72 | 6.84 | 6.99 | 7.14 |
| Izmit | 2_3 | 18 | 39.1 | 6.88 | 6.86 | 6.97 | 7.12 | 7.27 |
| Izmit | 1 | 18 | 24.7 | 6.68 | 6.63 | 6.75 | 6.90 | 7.05 |
| Izmit | 3+2_1 | 18 | 86.2 | 7.23 | 7.29 | 7.36 | 7.51 | 7.66 |
| Izmit | 2_1+2_2 | 18 | 81.8 | 7.21 | 7.26 | 7.34 | 7.49 | 7.64 |
| Izmit | 2_2+2_3 | 18 | 69.3 | 7.14 | 7.17 | 7.25 | 7.40 | 7.55 |
| Izmit | 2_3+1 | 18 | 63.8 | 7.10 | 7.13 | 7.21 | 7.36 | 7.51 |
| Izmit | 3+2_1+2_2 | 18 | 116.4 | 7.37 | 7.45 | 7.51 | 7.66 | 7.81 |
| Izmit | 2_1+2_2+2_3 | 18 | 120.9 | 7.38 | 7.47 | 7.53 | 7.68 | 7.83 |
| Izmit | 2_2+2_3+1 | 18 | 94 | 7.27 | 7.34 | 7.40 | 7.55 | 7.70 |
| Izmit | 3+2_1+2_2+2_3 | 18 | 155.5 | 7.50 | 7.61 | 7.65 | 7.80 | 7.95 |
| Izmit | 2_1+2_2+2_3+1 | 18 | 145.6 | 7.47 | 7.57 | 7.62 | 7.77 | 7.92 |
| Izmit | 3+2_1+2_2+2_3+1 | 18 | 180.2 | 7.56 | 7.69 | 7.73 | 7.88 | 8.03 |
| South Çınarcık | South Çınarcık | 18 | 39 | 6.86 | 6.88 | 6.97 | 7.12 | 7.27 |
| Background | - | 18 | - | - | | 6.5 | 6.80 | 7.1 |