# Peer review of "Planar Seismic Source Characterization Models Developed for Probabilistic Seismic Hazard Assessment of Istanbul"

_Natural Hazards and Earth System Sciences, 2017_

## Referee Comment (RC1) · Anonymous Referee #1 · 2 Jun 2017

This article exposes the development of a new hazard model for the city of Istanbul, Turkey. The model proposed mixes active faults and a background seismicity. The subject is pertinent and the overall article is well written and deserve to be published after some modifications are done: adding of a discussion about the slip-rate used in the model, the uncertainties and the output of the models, and improvement the figures.

Hereafter the list of issue concerning the article:

Main issues:

- The values of slip-rates used in this study are not referenced clearly enough and the uncertainties are not discussed enough.
  Here are a list of point concerning this issue:
  o *"slip rates should be participated* ». Comment: GPS does not provide slip rates for faults. Geodetic slip rates for major block-bounding structures are deduced from elastic block models. Suggestion: Geodetic slip rates deduced from elastic block models along the major block-bounding structures of the NAFZ. (McClusky et al. 2000; Meade et al., 2002; Reilinger et al., 2006). GPS data resolve left-lateral slip on the order of 25 mm/yr, with more than 80% being accommodated along the northern branch. Which is the reference providing this value of 80%.
  o *"slip rate of 19 mm/year is assigned to these segments of the northern strand and 6 mm/year is assigned to Geyve-Iznik Fault based on the values proposed by Stein et al. (1997) with slight modifications due to catalogue seismicity*." Why is there a need for modification of the slip-rate?
  o *"Since the contribution of Düzce Fault to the total slip is around 33% to 50% (Ayhan et al. 2001)*". What is the final contribution chosen here and why? Ayhan et al., 2001 states that analysis of GPS data suggest something different, that up to 10 mm/yr are accommodated on the Duzce-Karadere strand of the NAF [Ayhan et al., 1999].

    ➔ Suggestion: Please keep original reference when possible and explain how catalogue seismicity modifications led you to propose different slip rates for these two fault strands and could you please compare your slip rate estimates with more recent findings
    e.g. Ergintav, 2014 gives :
    ▪ for the Cynarcik Basin fault PIF: 10-15 mm/yr **vs** the 19 mm/yr with no uncertainty used in this study;
    ▪ for the Central Marmara region: < 2mm/yr **vs** the 19 mm/yr with no uncertainty used in this study.
    ➔ Suggestion: Table 1 please add original references used to estimate slip rates, add associated uncertainties and in the text justify your choice of slip rate with respect to the many alternative interpretations.

- The article targets to present "fully-documented and ready to use fault based SSC" (P1L18) which is a good way to share hazard model information. This approach deserves to be promoted in the seismic hazard community. Unfortunately, with this state of the paper, it is most possible to use the results for a reader in order to run a hazard calculation. The geometry of the faults and the background earthquake rates are provided in the supplements but the earthquake rate on faults is absent. Authors should provide these rates for the full logic tree described in this study. Furthermore, the authors should acknowledge the limitation of their

model and the uncertainties that remained unexplored in their logic tree (fault segmentation, fault geometries, slip-rate, scaling law used …) for future user to be able to use their work and run a complete and critical hazard assessment for the city of Istanbul.

- A logic tree is presented, with the exploration of several branches (b values, Mmax) but the results of the logic tree and the influence of each parameters is not exposed. A Discussion part should be added to the article in order to discuss the hazard model, to compare how it perform against the data (modeled seismic rate vs earthquake catalogue), discuss the issue of double counting, and to compare against the other seismic hazard model discussed in the intro. The limits of the models need to be clearly discussed as well. For example, the model allows multi-fault ruptures but the boundary of each system is based on the past earthquake rupture (Parson 2004) and the possibility of an earthquake passing from one system to another is not discussed.

- The issue of Mmax in the background zone should be discussed in greater detail: please refer to the extensive literature, UCERF3 in particular, for a more up to date discussion on this issue.

General comment:

- Why use the term "planar seismic source" instead of "fault source"?

Specific comments:

(format : page-line)

**1 Introduction**

2-1 please define "*floating earthquake*"

2-2 "*Parsons (2004) noted that the10 May 1556 (Ms=7.1), 2 September 1754 (M = 7.0), and 10 July 1894 (M = 7.0) earthquakes were located in the Çınarcık basin or on mapped normal faults in the southern parts of Marmara Sea*". Please change located with "were assigned location" as written by Parson 2014 " *1556 Ms _ 7.1, 2 September 1754 M _ 7.0, and 10 July 1894 M _ 7.0 earthquakes….were assigned locations in the Cinarcik basin or southern Sea of Marmara on mapped normal faults*"

2-30 "*preferred*" should be changed to *chosen*

3-1,2 please give references to the fault maps and satellite images

3-23 by "*seismic energy*", do you mean *moment budget*?

3-26 "*can be directly implemented*", as said before, there is a need for more information in order for the reader to implement the hazard model presented in this study.

**2 Fault Models, Rupture Systems, and Partitioning of Slip Rate**

The faults in the southern part of the Marmara Region are not included in the model. The background seismicity will in a way fix this problem latter on. In any case, the fact that only the Northern faults are taken into account should be clearly stated in the beginning of this part.

3-29 "previous large magnitude earthquakes" Parson 2004 needs to be cited here.

4-10 "…..its motion must be controlled by the motion of the Izmit Segment." What do you mean? Please justify and provide references.

4-29 The Geyve-Iznik fault is cited and a slip-rate is attributed but this fault is not kept in the final model. This fact should be clearly pointed in the text.

4-32 Here the author that the segment 1 of the Duzce fault is connected with the Izmit system. However, they cannot rupture together. Why so?

6-4 "*that act*" replaced by "*that can act*"

6-14 "*hence the most imminent seismic hazard to Istanbul and other cities*" this is true but since this paper is presenting a Poissonian model and not a time-dependent model, it is out of the scope of this study.

**4 Magnitude Recurrence Models – Seismic Moments**

What is the slip-rate chosen for a rupture scenario when faults don't have the same slip-rate (for example, S 2_2 and S 2_3 in S 2_2+2_3)

7-8 "*the catalog was assumed to be complete for 52 years for Mw $\leq$ 4.5 and Mw $\leq$ 5.0*" do you mean Mw >=4.5 or 5.0? Please be precise the on the completeness time for each magnitude range.

7-30 "Seismic sources generate varied sizes of earthquakes" change to "Seismic sources can generate various sizes of earthquakes"

8-6 "fault zones" do you mean individual faults? The GR distribution can work quite well for a fault zone if several faults are in this zone.

8-17 "*relative rates of small, moderate and large earthquakes*" The term MFD can be introduced here.

8-18 "*related to the rupture system*" As I understand, this MFD is attributed for each possible rupture of the model. A rupture system will be the sum of these MFD, something different from a Youngs and Coppersmith distribution.

9-5 Can "*magnitude PDF*" be replaced by MFD?

9-6 Why this choice of adding 0.25 and 0.5 to the Mmax define using Wells and Coppersmith 1994? Doesn't make the new Mmax not fitting the scaling law? Why not explore the uncertainty given by Well and Coppersmith or another scaling law in order to grasp the epistemic uncertainty?

Equation (4) – what is value of $\mu$ used in this study and based on which data?

9-20 is the *moment-balancing* the same for all the branches of the logic tree? What is the branch presented in figure 4?

9-22 the "*best fit*" between the rate in the catalogue and the weighted average is defined in which way? It seems that the fit with the smaller magnitude is preferred according to fig 4 because of the large uncertainty on the rate of large magnitude earthquake. Why the authors didn't choose to use an historical earthquake catalogue in order to improve the estimation of the rate of larger earthquakes?

9-29 higher weight is attributed to single rupture than to multiple fault rupture. What are the basis for this assumption since the distribution used (Youngs and Coppersmith) already predicts more small

magnitude earthquakes than large ones? Is this argument stronger than the fit to the data in the weight determination?

**5 Background Zone – Smoothed Seismicity**

10-4 define "*not associated*". What is the size of the buffer zone? And why? Please state whether the background zone and the fault sources should be superposed in the PSHA calculations. (Not clear in figure 5)

10-7 "*distinctive zones of seismicity are not observed*": what do you mean by this?

10-21 "*no active fault has been reported*". Faults in the vicinity of Istanbul have been described in other studies. See Diao et al 2016 (Secondary Fault Activity of the North Anatolian Fault near Avcilar, Southwest of Istanbul: Evidence from SAR Interferometry Observations).

**Discussion part is missing**

**6 Conclusion**

10-30 "*previous SSC models*": a comparison on the modelled rate will improve the quality of the article.

11-12 "*can be directly implemented*" I agree that sharing a properly documented hazard model is a goal that more PSHA study should aim too. For this paper, information is still needed in order to accomplish that goal.

11-18 this interesting comparison with other model could be done in the discussion part in greater depth.

**References**

please have a look at the format for this journal. I think the "&…" is not accepted and doi should be provided when available.

**Figures**

Figure 1

A color code for each rupture system could be used. The full name of each rupture system should be indicated on the map to help the reader. What is the number between brackets?

Figure 4

Lines and Points are too thick and make the figure difficult to read. Please indicate which branch of the logic tree is exposed here (Mmax, b value). What do the blue and purple colors correspond too? This figure deserve to be explain more clearly. Please unify the style of the four figures.

Figure 5

Please complete the legend: scale of the color scale, red dots.

Choose better color code for the faults and add the names. (Linked with the rupture system for example)

**Tables**

Table 3

"*Logic tree for style of faulting*". This is not a logic tree since the style of faulting is not an epistemic but an aleatory uncertainty. I needs to be specified that the table refers to the background zone.

Table 4

The rupture scenario is not an epistemic uncertainty but an aleatory one.

---

## Referee Comment (RC2) · Anonymous Referee #2 · 3 Jul 2017

This is a technically-solid, well-documented paper describing the implementation of a seismic source model for the North Anatolian Fault Zone.

The paper is not a research paper, and therefore does not really attempt to advance new ideas or change the way the earthquake process is understood in the region. Instead, it simply describes a segmented seismic source model and the calibration of parameters of interest (e.g. Gutenberg-Richter A+B values) to the faults.

Whether this is appropriate for this journal or not, I cannot say definitively. It would have been nice to see a little more scientific research.

However, the work that is done is of good quality, quite well documented and no doubt

of use and interest to the community.

My only technical concerns are that the B values estimated for the faults are quite low ($\sim$0.7). This is may be due to catalog completeness issues, or overly aggressive declustering that removes too many events. Though the methods used to decluster the catalog are mentioned, there are no statistics on the number or percentage of events removed or other information that would help with this sort of diagnostics.

Alternately, it is possible that the NAFZ does have a very low B value. This would be quite notable, and worthy of more scientific investigation. I am not a regional expert so I cannot comment on this directly, but it is necessary to discuss in the manuscript.

---

## Author Comment (AC1) · 18 Aug 2017

Dear Editor,

Enclosed please find the revision of Manuscript ID: NHESS-2017-113 entitled "**Planar Seismic Source Characterization Models Developed for Probabilistic Seismic Hazard Assessment of Istanbul**". We appreciate your time and efforts during the review process. We are also thankful to the reviewers for valuable and constructive comments and for encouraging statements about the manuscript. In the revision, we have taken into account all the comments and made changes accordingly. Details of the actions taken regarding the comments and edits are provided below (all page, line, and figure numbers are given according to revised annotated manuscript).

**Reviewer #1 – Main Issues**

> *"This article exposes the development of a new hazard model for the city of Istanbul, Turkey. The model proposed mixes active faults and background seismicity. The subject is pertinent and the overall article is well-written and deserves to be published after some modifications are done: adding of a discussion about the slip-rate used in the model, the uncertainties and the output of the models, and improvement the figures."*

We thank the reviewer for the encouraging statements. Details of the changes we made are summarized below.

> 1. *GPS does not provide slip rates for faults. Geodetic slip rates for major block-bounding structures are deduced from elastic block models.*

As suggested by the reviewer, mentioned sentence is changed as follows: *"Past studies based on GPS measurements (McClusky et al. 2000; Meade et al., 2002; Armijo et al., 2002; Reilinger et al., 2006) suggest a 22 ± 3 mm/yr dextral motion along the major block-bounding structures of the NAFZ, with more than 80% being accommodated along the northern branch."* Page 5- Lines 1-4.

> 2. *"Slip rate of 19 mm/year is assigned to these segments of the northern strand and 6 mm/year is assigned to Geyve-Iznik Fault based on the values proposed by Stein et al. (1997) with slight modifications due to catalogue seismicity."* Why is there a need for modification of the slip-rate?

Mentioned sentence was not clear enough to explain the applied procedure. In the "moment-balanced" seismic source models, the magnitude recurrence model parameters given in Eq. 4 (Page 10) such as the annual slip rate, b-value, etc. are tested for consistency with the rate of earthquakes associated with the rupture system. These graphs for all rupture systems are given in Figure 4. Eq. 1 shows that the annual slip rate directly increases the accumulated seismic moment; therefore, increasing the annual slip rate moves the red broken lines in Figure 4 upwards. The slip rate participation among the northern strand of NAFZ and Geyve-Iznik fault was given as 16 mm/yr and 9 mm/yr in Stein et al. (1997). However, we achieved a better fit with the associated seismicity of Izmit rupture system by increasing the share of the northern strand of NAFZ to 19 mm/yr. This value is also in good agreement with the annual slip rate given in Murru et al. (2016): they have adopted 20±2 mm/yr based on the proposals of Flerit et al. (2003) and Ergintav et al. (2014). We changed that sentence to clarify this issue (Page 6, Lines 10-14).

> 3. *"Since the contribution of Düzce Fault to the total slip is around 33% to 50% (Ayhan et al. 2001)". What is the final contribution chosen here and why? Ayhan et al., 2001 states that analysis of GPS data suggest something different, that up to 10 mm/yr are accommodated on the Duzce-Karadere strand of the NAF [Ayhan et al., 1999]. Please keep original reference when possible and explain how catalogue seismicity modifications led you to propose different slip rates for these two fault strands.*

As mentioned by the reviewer, Ayhan et al. (2001) suggested that up to 10 mm/yr of the motion is accommodated on the Düzce-Karadere strand of the NAF. We also utilized the same annual slip rate of 10 mm/yr for Düzce_1, Düzce_2 and Karadere segments without any modifications based on the catalogue. Related text in Page 6 (Lines 18-20) is now updated, citing Ayhan et al (2001).

4. *Could you please compare your slip rate estimates with more recent findings? E.g. Ergintav, 2014 gives 10-15 mm/yr for the Çınarcık Basin fault PIF vs the 19 mm/yr with no uncertainty used in this study; < 2mm/yr for the Central Marmara region: vs the 19 mm/yr with no uncertainty used in this study.*

The slip rate estimate given in Ergintav et al. (2014) for the Prince Island Fault and Çınarcık Basin is 15±2 mm/year (page#5784 of the original reference). Murru et al. (2016) distributed the annual slip rate of 17 mm/year among two parallel branches in this zone; 14±2 mm/year for Çınarcık segment and 3±1 mm/year for the South Çınarcık segment based on the recent works of Ergintav et al. (2014) and Hergert and Heidbach (2010). Therefore, the slip rate value that we have used on the horizontal plane is identical to these recent estimates (Figure 1d). The slip rate given for the Central Marmara Fault by Ergintav et al. (2014) (2 mm/year) is unusually low compared to the previous estimates and may be suffering from the sparsity of the network and GPS coverage on the north shores of Marmara Sea as mentioned by the authors (page#5786). For this rupture system, the annual slip rate we adopted (19±2 mm/year) is in good agreement with the proposal of Murru et al. (2016) (18±2 mm/year) and with the seismicity rates based on instrumental earthquake catalogue (please refer to Figure 4b). The text given here is added to Pages 6-7, Lines (31-10).

5. *Table 1 please add original references used to estimate slip rates, add associated uncertainties and in the text justify your choice of slip rate with respect to the many alternative interpretations.*

Table 1 is modified as suggested by the reviewer. New Table 1 now includes the references for adopted slip rates and uncertainty in the published slip rate values. Additionally, we modified the SSC logic tree to include the epistemic uncertainty in the slip rates and changed the caption of Table 1 accordingly.

6. *The article targets to present "fully-documented and ready to use fault based SSC" (P1L18) which is a good way to share hazard model information. This approach deserves to be promoted in the seismic hazard community. Unfortunately, with this state of the paper, it is most possible to use the results for a reader in order to run a hazard calculation. The geometry of the faults and the background earthquake rates are provided in the supplements but the earthquake rate on faults is absent. Authors should provide these rates for the full logic tree described in this study.*

Thank you for supporting the open access policy for the seismic source models. Typically, the hazard codes do not need the earthquake rates on the fault. The magnitude PDF among the predefined models in the code is selected (in our case this is the Youngs and Coppersmith (1985) composite model), magnitude PDF parameters should be entered (in our case the b-value, $M_{min}$, and $M_{char}$) and the earthquake rates are implicitly calculated by the hazard code based on the provided logic tree for each seismic source (in our case for each rupture source). Nevertheless, following sentences are added to the manuscript and the earthquake rates are now provided in the electronic supplement. *"The hazard analyst can incorporate the full rupture model and the complete logic tree provided in this manuscript to most of the available hazard codes without explicitly calculating the earthquake rates. In case that the earthquake rate has to be incorporated to the hazard code; the earthquake rates for each branch of the logic tree given in Electronic Supplement#3 can be used."* (Page 14, Lines 16-19).

7. *Furthermore, the authors should acknowledge the limitation of their model and the uncertainties that remained unexplored in their logic tree (fault segmentation, fault geometries, slip-rate, scaling law used…) for future user to be able to use their work and run a complete and critical hazard assessment for the city of Istanbul.*

8. *A logic tree is presented, with the exploration of several branches (b values, Mmax) but the results of the logic tree and the influence of each parameters is not exposed. A Discussion part should be added to the article in order to discuss the hazard model, to compare how it perform against the data (modeled seismic rate vs earthquake catalogue), discuss the issue of double counting, and to compare against the other seismic hazard model discussed in the intro. The limits of the models need to be clearly discussed as well. For example, the model allows multi-fault ruptures but the boundary of each system is based on the past*

*earthquake rupture (Parson 2004) and the possibility of an earthquake passing from one system to another is not discussed.*

Following the suggestions of the reviewer, we added a new Discussion section that deliberates the SSC model parameters and the epistemic uncertainty of the model based on the comparison of the source model fractals of each rupture source with the observed rates of associated earthquakes (Section #6). We added a paragraph to the newly introduced Section#6 that discusses the uncertainties remained unexplored in the provided logic tree. We also shortly discussed the reason why the fault-to-fault rupture concept of UCERF3 is not utilized in the proposed model at the end of this section.

Additionally, we added the following sentence to the main text: "*During the calculations of the smoothed seismicity rates, the earthquakes in buffer zones are not included in smoothing (and not double-counted). The buffer zones are only used to "associate" the earthquakes with the fault zones and collapse the earthquakes to the vertical fault planes.* (Page 12 – Lines 18-21)".

9. *The issue of Mmax in the background zone should be discussed in greater detail: please refer to the extensive literature, UCERF3 in particular, for a more up to date discussion on this issue.*

We appreciate the suggestion. Moschetti et al. (2015) mentioned that the development of the maximum magnitude ($M_{max}$) model for shallow crustal seismicity in the Western United States benefits from the large set of regional earthquake magnitudes from the historical and paleoseismic records; however, the background seismicity model accounts for earthquake ruptures on unknown faults; therefore, the $M_{max}$ distribution must reflect the range of possible magnitudes for these earthquakes. We adopted a similar approach using the fault segments of the southern strand of NAFZ documented in Murru et al. (2016) and calculated the characteristic magnitude for each segment with Wells and Coppersmith (1994) magnitude-rupture area relation. Based on the estimations of characteristic magnitude of earthquakes that may occur on the southern strand of NAFZ, the logic tree for $M_{max}$ of the background zone is modified (Table 6). Related discussion is added to Page 12, Lines 25-32.

10. *Why use the term "planar seismic source" instead of "fault source"?*

Planar seismic source is preferred to emphasize the third dimension of the fault plane.

**Reviewer #1 – Specific Comments:**

Language edits in all sections are acknowledged. We are indebted for the careful grammar review. Some of the issues pointed out by the reviewer are resolved by adding further explanations throughout the text (please refer to the annotated manuscript). We would like to add a few remarks for addressing some of the specific comments:

1. The references to the fault maps and satellite images used by Gülerce and Ocak (2013) are provided in the original reference; therefore, the details are not elaborated here due to page limitations.
2. 4-32 Here the author that the segment 1 of the Duzce fault is connected with the Izmit system. However, they cannot rupture together. Why so?

In 1999 earthquakes, these two fault systems (Kocaeli and Düzce) were ruptured in two different episodes. A possible explanation of the separate ruptures in different episodes would be the development of the restraining bend along Karadere Segment, which probably locked up the eastern termination of Izmit rupture. Harris et al. (2002) proposed that the rupture of 1999 İzmit earthquake was stopped by a step-over at its eastern end (Mignan et al., 2015). Within the scope of this study, we believe that it is safe to assume the same rupture pattern of 1999 earthquakes based on current information. However, we added the sentences above to the manuscript (Page 4, Lines 24-29).

3. 9-6 Why this choice of adding 0.25 and 0.5 to the $M_{max}$ define using Wells and Coppersmith 1994? Doesn't make the new $M_{max}$ not fitting the scaling law? Why not explore the uncertainty given by Well and Coppersmith or another scaling law in order to grasp the epistemic uncertainty?

We thank the reviewer for pointing that out. We changed the structure of the maximum magnitude logic tree using two different magnitude scaling relations proposed by Wells and Coppersmith (1994) and Hanks and Bakun (2014). The $M_{char}$ values calculated using both equations are quite close to each other and the absolute value of the difference is smaller than 0.13 in magnitude units. To grasp the epistemic uncertainty, average of the $M_{char}$ value from both scaling laws are utilized in the center of the logic tree with 50% weight and both the $M_{char}$ -0.15 and $M_{char}$ +0.15 values are included by assigning 25% weight (Table 6).

4. 9-20 is the moment-balancing the same for all the branches of the logic tree? What is the branch presented in figure 4?

No, it is not the same for all branches of the logic tree. We modified the caption of Figure 4 to indicate the branch of the logic tree presented in each part figure.

5. 9-22 the "best fit" between the rate in the catalogue and the weighted average is defined in which way? It seems that the fit with the smaller magnitude is preferred according to fig 4 because of the large uncertainty on the rate of large magnitude earthquake. Why the authors didn't choose to use an historical earthquake catalogue in order to improve the estimation of the rate of larger earthquakes?

The best fit between the rates of the events in the instrumental catalogue and the weighted average of the magnitude recurrence model is achieved by visual interpretation. To achieve a good fit, the seismic source modeler needs to understand the contribution of the magnitude recurrence model parameters to the red broken line in different magnitude ranges. For example, the b-value significantly affects the small magnitude portion of the curve since the Youngs and Coppersmith (1985) magnitude PDF is utilized. Please remind that the b-value is calculated based on the same catalogue but for a larger region when compared to the buffer zone around the fault. Defining a large number of sub-segments for a rupture system also increases the cumulative rate of small magnitude events. The good fit in the small magnitude range of Figure 4 shows that: i) the b-value calculated using the larger zone is compatible with the seismicity associated with the planar source, ii) utilized segmentation model is consistent with the relative rates of small-to-moderate and large events, and iii) annual slip rate is compatible with the seismicity over the fault. As the reviewer mentioned, the large magnitude rates are poorly constrained since the catalogue used herein only covers 110 years and that time span is obviously shorter than the recurrence rate for the large magnitude event. Hecker et al. (2013) explains that by the low rates of the large magnitude events: *"rates of large-magnitude earthquakes on individual faults are so low that the historical record is not long enough to test this part of the distribution"* and suggest using the *"inter-event variability of surface-rupturing displacement at a point as derived from geologic data sets"* to test the upper part of the earthquake-magnitude distribution. Discussion given above is added to the manuscript (Page 11, Lines 15-27).

6. 9-29 higher weight is attributed to single rupture than to multiple fault rupture. What is the basis for this assumption since the distribution used (Youngs and Coppersmith) already predicts more small magnitude earthquakes than large ones? Is this argument stronger than the fit to the data in the weight determination?

Both truncated exponential model and the Youngs and Coppersmith (1985) model assumes more small magnitude events than the large magnitude events. The difference lies in the relative rates of small-to-moderate and large magnitude earthquakes (for further details please refer to Hecker et al., 2013 and Gülerce and Vakilinezhad, 2015). However, the ratio of these rates is the same for the single-segment rupture "source" and for the multiple-segment rupture "source" and this ratio is irrelevant with the weights assigned to the rupture "scenarios". Higher weights attributed to the single-segment rupture scenarios than the multiple-segment rupture scenarios reflect the preference of the seismic source modeler in addition to the agreement with the associated seismicity. As Figure 4 implies, this preference did not contradict with the cumulative rates of earthquakes associated with each rupture system.

7. 10-4 define "not associated". What is the size of the buffer zone? And why? Please state whether the background zone and the fault sources should be superposed in the PSHA calculations. (Not clear in figure 5)

Size of the buffer zone is 7 km in each side of the fault line based on the visual interpretation of the spatial distribution of the earthquakes around the fault lines. We assumed that the earthquakes within the buffer zones are "associated" with the fault and the ones that are outside of the buffers are "not associated". Following sentences are added for clarification (Page 12 – Lines 18-21): *"During the calculations of the smoothed seismicity rates, the earthquakes in buffer zones are not included in smoothing (and not double-counted). The buffer zones are only used to "associate" the earthquakes with the fault zones and to collapse the earthquakes to the vertical fault planes. Therefore, the background source and the fault sources can be superposed in the PSHA calculations."*

8. 10-21 "no active fault has been reported". Faults in the vicinity of Istanbul have been described in other studies. See Diao et al 2016 (Secondary Fault Activity of the North Anatolian Fault near Avcilar, Southwest of Istanbul: Evidence from SAR Interferometry Observations).

Greater Istanbul Municipality had conducted a trench study on the KL Fault of Diao (2016) in order to verify its recent activity; however, they found no evidence of Holocene activity. Avcılar region is dominated by active and extensive landslides and surface creep activities as Diao et al. (2016) suspected.

9. 10-30 "previous SSC models": a comparison on the modelled rate will improve the quality of the article.
10. 11-18 this interesting comparison with other model could be done in the discussion part in greater depth.

We appreciate the comments and understand the importance of the comparison of the earthquake rates proposed in here with the previous literature. Unfortunately, previous publications did not provide enough information on earthquake rates for doing this comparison. A similar statement is added to the discussion section to underline the importance of open-access seismic source models in PSHA.

11. Figure 1: A color code for each rupture system could be used. The full name of each rupture system should be indicated on the map to help the reader. What is the number between brackets?

The numbers between brackets were segments lengths. Figure 1 is modified as suggested and the numbers (segment lengths) are deleted for clarity (instead the segment lengths are given in the Table 1) and a color code for each rupture system is introduced.

**Reviewer #2 – General comments.**

*"This is a technically-solid, well-documented paper describing the implementation of a seismic source model for the North Anatolian Fault Zone. The paper is not a research paper, and therefore does not really attempt to advance new ideas or change the way the earthquake process is understood in the region. Instead, it simply describes a segmented seismic source model and the calibration of parameters of interest (e.g. Gutenberg-Richter A+B values) to the faults. Whether this is appropriate for this journal or not, I cannot say definitively. It would have been nice to see a little more scientific research. However, the work that is done is of good quality, quite well documented and no doubt of use and interest to the community."*

We thank the reviewer for the encouraging statements. The state-of-the-art in seismic hazard assessment and seismic source characterization models are generally published in consultancy reports and typically not easily accessible for the earthquake engineering practitioners. Abrahamson (2000) proposed that one of the sources of the problems leading to the large variability in the seismic hazard practice is the lack of well-written, easy to understand papers on the topic of seismic hazard assessment. With the help of the review comments, the manuscript is significantly improved; therefore, we hope that the reviewer would see the scientific and/or practical contribution of the updated manuscript.

*My only technical concerns are that the B values estimated for the faults are quite low (≈0.7). This is may be due to catalog completeness issues, or overly aggressive declustering that removes too many events. Though the methods used to decluster the catalog are mentioned, there are no statistics on the number or percentage of events removed or other information that would help with this sort of diagnostics. Alternately, it is possible that the NAFZ does have a very low B value. This would be quite notable, and worthy of more scientific*

*investigation. I am not a regional expert so I cannot comment on this directly, but it is necessary to discuss in the manuscript.*

The b-value estimated for Zones 1-3 varies between 0.68-0.76. We understand that the estimated values are relatively small when compared to the b-values estimated for large zones (b≈1); however, our findings are consistent with the current literature. Şeşetyan et al. (2016) provided a thorough analysis of the b-value for the whole Turkish territory and proposed that b=0.77 for Central Marmara region and b=0.67 for North Anatolian Fault Zone (please refer to Figure 15 of Şeşetyan et al., 2016). The catalogue completeness intervals used in Şeşetyan et al. (2016) and in this study for 4.7<M<5.7 earthquakes are exactly the same; therefore, we do not expect that the b-values estimated here depends on the catalogue completeness intervals utilized in this study. The small differences in the b-values proposed by Şeşetyan et al. (2016) and the b-values estimated in this study due to the geometry of the selected zones and the differences in the compiled catalogues. In addition, the b-value used by Moschetti et al. (2015) for Western United States (b=0.8) is not very different than our estimates. Discussion given above is added to the manuscript (Pages 8-9, Lines 29-2).

The reviewer suggested that the estimated b-value might be affected from the aggressive declustering that removes too many events. This issue is thoroughly discussed in Güner et al. (2015) and Azak et al. (2017), showing that the declustering methodology utilized in this study (Reasenberg, 1985) results in higher estimates of the b-value when compared to the other declustering methods. We provided additional details on the declustering at Page 7 (line 31).

Finally, we would like to underline that the b-value only controls 6% of the released seismic moment by the exponential tail of the implemented composite magnitude PDF (Youngs and Coppersmith, 1985), therefore it has no substantial effect on the hazard (Gülerce and Vakilinezhad, 2015).

**References given here (but not included in the manuscript):**

1) Abrahamson, N. A.: State-of-the-practice in seismic hazard evaluation, ISRM International Symposium, 2000.
2) Eroğlu Azak, T., Kalafat, D., Şeşetyan, K., Demircioğlu, M. B.: Effects of seismic declustering on seismic hazard assessment: a sensitivity study using the Turkish earthquake catalogue Bulletin of Earthquake Engineering (online first), DOI 10.1007/s10518-017-0174-y, 2017.
3) Diao, F., Walter, T. R., Minati, F., Wang, R., Costantini, M., Ergintav, S., et al.: Secondary Fault Activity of the North Anatolian Fault near Avcilar, Southwest of Istanbul: Evidence from SAR Interferometry Observations, Remote Sensing, 8(10), 846, doi:10.3390/rs8100846, 2016.
4) Güner, B., Menekşe, A., Özacar, A. A., and Gülerce, Z: Kuzey Anadolu ve Doğu Anadolu Fay Zonu için Deprem Tekrarlanma Parametrelerinin Belirlenmesi, 3. Türkiye Deprem Mühendisliği ve Sismoloji Konferansı Bildiri Kitabı, 14-16 Ekim 2015, İzmir (in Turkish).

---

## Author Comment (AC2) · 21 Aug 2017

The revised and annotated manuscript and the electronic supplements are attached.

Please also note the supplement to this comment:
https://www.nat-hazards-earth-syst-sci-discuss.net/nhess-2017-113/nhess-2017-113-AC2-supplement.zip
* * *

---

## Referee Report (RR1)

The authors have improved their article to a great extent and provided the necessary information to run their model in a PSHA calculation. The Excel files are very much appreciated. This article can be published after some corrections are made and some figures are improved.

**General remarks**:

Many references cited in the text are missing :

P2 : Saroglu et al 1992 ; P4: Mert et al 2016, Laigle et al 2008, Harris et al 2002, Mignan et al 2015 ; P7 : Sesetyan er al 2016 ; P8 : Moschetti et al 2015 ; P9: Hanks and Bakun 2014; P10: Brodsky et al 2000, Field et al 2009; P13: Field et al 2013

Discussion of the results: Instead of plotting the distribution for each hypothesis (single faults or complex rupture), I suggest to plot the distribution of the final result of each model after the weighting in order to check the validity of the model against the data. This will provide the reader and (the user) a better visualization of the epistemic uncertainty affecting the model than separating the mean single segment ruptures and multi segment ruptures (such as presented in the present version of the article).

See here after an example for the Duzce fault made using the supplementary material.

[Figure]

Figure: Cumulative earthquake rate for the Duzce fault system. Black squares are the rate of the catalogue. Red line: weighted mean of the logic tree. Dotted line: + and – sigma.

For these results to be "ready to use", the supplementary information should describe the hypotheses behind each branch. A system of branch number can be used; for example:

|  | Branch 1 | Branch 2 | Branch 3 |
| --- | --- | --- | --- |
| b value 1 | 1 | 0 | … |
| b value 2 | 0 | 1 | |
| b value 3 | 0 | 0 | |
| | | | |
| Mmax1 | 1 | 1 | |
| Mmax2 | 0 | 0 | |
| Mmax3 | 0 | 0 | |
| | | | |
| slip-rate 1 | 1 | 1 | |
| slip-rate 2 | 0 | 0 | |
| slip-rate 3 | 0 | 0 | |

**Hereafter are some more specific remarks**:

P1L27 "Parson collected" should be change to "Parson proposed" in my opinion.

P3L2 "more complicated" should be changed to "more detailed".

P3L9 Rephrase the sentence. The "however" can be removed.

P4L2 Murru et al 2016 uses a dip of 60° for the South Cinarcik fault. Please cite the source of this near vertical dip.

P4L2 Please explain here that the model developed here is only valid for PSHA in Istanbul, not for the whole Marmara region.

P5L29 Missing citation of Sengor 2014.

P5L31 Why Hergert and Heibbach 2010; and Ergintav 2014 are not cited?

P6L4 The authors here claim to achieve a better fit by changing the slip-rate on the northern strand of the NAF. Please provide reference or a more detailed analysis in order to justify a change of the slip-rate of a major fault.

P6L24 "proposal" needs to be changed. Murru et al 2016 are not calculating or measuring slip-rates, please cite the references in Murru et al 2016 that led them to set this value of slip-rate.

P6L31 Herget et al needs to be changed in Hergert and Heidbach.

P8L21 Please provide a reference for the truncated exponential equation.

P10L4 The weighted average of the slip-rate is an important assumption of the methodology, I think it deserve to be written as an equation to help the reader understand.

P10L25 "utilized" => "used"

P12L8 The value of Mmax used in the background deserves to be clearly indicated in the text.

P12L10 please give a reference for the value of 18 km used as a seismogenic depth.

P12L14 The authors claim to explore the uncertainties affecting the slip-rate and the Mmax. Please describe more where these uncertainties are explored and how the results are affected. The supplementary material unfortunately doesn't contain any information concerning the logic tree.

P12L31 Please provide references of ongoing paleoseismic research.

P12L32 the last two sentences of this paragraph should be in the introduction as they are a reason why you had to develop your alternative approach to UCERF3.

P13L15 Unsure if it is necessary to precise that the table document your model. It's self-explanatory in the article.

P13L19 The fact that this model is only valid for calculations in Istanbul needs to be stressed throughout the article.

Figure 1 : please place a) b) c) d) e) on the figure. The last sentence of the caption is not necessary.

Figure 4 needs to be improved. Use finer lines and points (such as in Figure 6) in order to let the reader see the value of each single point.

Figure 6 in the caption; explain what are a) b) c) d). Be careful when using excel figure with a logarithmic scale, the last point tends to be cut. For this reason, it is not possible to read the standard deviation of the rate for the largest magnitudes.

**Supplement** :

Background rates : please give the rates for each b value of the logic tree.

Fault rates : please give and explanation of the hypothesis of the branches.

---

## Author Response (AR2)

Dear Editor,

Enclosed please find the revision of Manuscript ID: NHESS-2017-113 entitled "**Planar Seismic Source Characterization Models Developed for Probabilistic Seismic Hazard Assessment of Istanbul**". We appreciate your time and efforts during the review process. We are also thankful to the reviewer for valuable and constructive comments about the revised manuscript.

In this revision, the referee report includes three general remarks:

1) **Missing references:** We thank the reviewer for the careful review of the manuscript. All of these references were added to the annotated manuscript during the previous review cycle but somehow were not included in the clean version. Now all of them are placed in the reference list.

2) **Discussion of the results based on Figure 6:** Current version of Figure 6 compares the source model fractals with the cumulative rate of the associated seismicity with the rupture system to evaluate the epistemic uncertainty included in the source model. The reviewer suggested adding the earthquake rates given in Supplement#2 on this figure to provide a better visualization of the epistemic uncertainty in the model. We appreciate the suggestion and the efforts for providing the example figure in the referee report. However, the example figure given in the report does not reflect the complete set of curves to be added to Figure 6. Since the Düzce, Ganos/Saros, and Central Marmara rupture systems include two segments and three rupture sources (S1, S2, S1+S2) and each rupture source have 27 parameter combinations; 3x27=81 lines have to be added to Figure 6b, 6c, and 6d, individually. For Izmit rupture system with 15 rupture sources (and again 27 parameter combinations), the curves to be added on Figure 6a are over four hundred. We think that adding these set of lines will not improve the visualization of the epistemic uncertainty included in the model and prefer to use the fractals to represent these large sets of curves.

3) **Parameter combinations for logic tree branches:** The reviewer recommended adding a coding system (with branch numbers) that combines the logic tree branches with the earthquake rates given in the supplement. We introduced this system in the updated Supplement#2.

In addition to the general remarks, the reviewer provided a list of specific remarks. We implemented all suggested changes, except for moving the discussion related to UCERF3 to the introduction section. We think that the discussion section is more appropriate for this information considering the context of the paper.